# Paralogs of Slitrk cell adhesion molecules configure excitatory synapse specificity via distinct cellular mechanisms

Dongwook Kim[1,2☯], Byeongchan Kim[1☯], Jinhu Kim[1,2☯], Na-Young Seo[1,3], Hyeonho Kim[1,2], Kyung Ah Han[1,2¤], Jubeen Yoon[4,5,6], Christian P. Macks[6], Joris de Wit[7,8], Chang Ho Sohn[6], Kea Joo Lee[3], Ji Won Um[1,2], Jaewon Ko [1,2]*

1 Department of Brain Sciences, Daegu Gyeongbuk Institute of Science and Technology (DGIST), Daegu, Korea, 2 Center for Synapse Diversity and Specificity, DGIST, Daegu, Korea, 3 Neural Circuits Research Group, Korea Brain Research Institute (KBRI), Daegu, Korea, 4 Department of NanoBiomedical Engineering (NanoBME), Advanced Science Institute, Yonsei University, Seoul, Korea, 5 Center for Nanomedicine, Institute for Basic Science, Seoul, Korea, 6 Graduate School of Medical Science and Engineering, Korea Advanced Institute of Science and Technology (KAIST), Daejeon, Korea, 7 VIB Center for Brain & Disease Research, Leuven, Belgium, 8 KU Leuven, Department of Neurosciences, Leuven Brain Institute, Leuven, Belgium

☯ These authors contributed equally to this work.
¤ Current address: Department of Anatomy and Cell Biology, College of Medicine, Chungnam National University, Daejeon, Korea
* jaewonko@dgist.ac.kr

## Abstract

Vertebrate neural circuit properties are shaped by synaptic cell adhesion molecules (CAMs). CAMs often have multiple paralogs but the possible redundancy of such paralogs remains underexplored. Using circuit-specific conditional knockout (cKO) mice deficient for Slitrk1 and Slitrk2, we show that these paralogs lack specific laminar expression in mature hippocampal neurons but divergently guide the specificity of neural circuits in distinct hippocampal subfields. Slitrk1 and Slitrk2 regulate distinct facets of excitatory synaptic properties in a microcircuit-dependent manner through binding to LAR-RPTPs, and additionally in the case of Slitrk2, through binding to PDZ domain-containing proteins and TrkB. Analyses of Slitrk2 V89M knock-in mice revealed that this schizophrenia-associated substitution acts uniquely as a loss-of-function mutation in some microcircuits to impair excitatory synaptic transmission, asynchronous release, and spatial reference memory. These findings demonstrate that even structurally and biochemically similar synaptic CAMs can play distinct roles in specifying neural circuit architecture.

## Introduction

Synapses are basic elementary units of neural information transfer that connect neurons into specifically wired neural circuits [1,2]. During development, synapses undergo seemingly distinct stages of biological processes that are shaped by diverse

**Data availability statement:** All relevant data are within the paper and its Supporting information file.

**Funding:** This work was supported by the National Creative Research Initiative Program of the Ministry of Science and ICT (https://www.msit.go/kr/eng; RS-2022-NR070708 to J.Ko), the National Research Foundation of Korea (NRF; https://www.nrf.re.kr/eng/main) funded by the Ministry of Science and ICT (RS-2023-00207834, 2022M3E58081184, and 2023R1A2C2002535 to J.W.U.; RS-2021-NR061982 to K.A.H.; RS-2024-00339642 to H.K.; and 2021R1C1C1004092 to C.H.S.), the KBRI basic research program (https://www.kbri.re.kr; 24-BR-01-03 to K.J.L.), ERC Starting Grant (https://erc.europa.eu; #311083 to J.d.W.), and FWO Project (https://fwo.be/en; G0C4518N to J.d.W.). The funders had no role in the study design, data collection and analysis, decision to publish, or preparation of the manuscript.

**Competing interests:** The authors declare that no competing interests exist.

**Abbreviations:** AAVs, Adeno-associated viruses; aCSF, artificial cerebrospinal fluid; ANOVA, one-way analysis of variance; AP, action potential; AZ, active zone; CAMs, cell adhesion molecules; Cas9, CRISPR-associated protein 9; CRISPR, clustered regularly interspaced short palindromic repeats; Cre, Cre recombinase; DCZ, Deschlorocloazapine; DMSO, dimethyl sulfoxide; DG, dentate gyrus; eEPSCs, evoked excitatory postsynaptic currents; eMAP, epitope-preserving magnified analysis of proteome; ES, embryonic stem; FITC, fluorescein isothiocyanate; FLRT2, fibronectin leucine rich transmembrane protein 2; IACUC, Institutional Animal Care and Use Committee; KD, knockdown; LAR-RPTPs, leukocyte common antigen related receptor protein tyrosine phosphatases; Lphn2, latrophilin-2; LRR, leucine-rich repeat; MML, medial molecular layer; MPP, medial perforant path; NDS, normal donkey serum; NGL-1, netrin-G ligand 1; Nlgn2, neuroligin-2; NMDAR, N-methyl-D-aspartate receptor; OsO4, osmium tetroxide; OML, outer molecular layer; PFA, paraformaldehyde; PBS, phosphate-buffered saline; PPRs, paired pulse ratios;

actions of synaptic cell adhesion molecules (CAMs). Synaptic CAMs are involved in dictating synapse assembly, specification of synaptic properties, synaptic plasticity, and even synapse elimination. Moreover, synaptic CAMs are intimately linked to both presynaptic specializations and postsynaptic machineries, where they serve to coordinate the precise alignment between pre- and post-synaptic compartments and potentially compart nanodomain architecture [2,3]. A growing appreciation of the central importance of CAM functions in specifying neural circuits has motivated extensive investigations of various CAMs, and identification of candidate synaptic CAMs has proceeded apace; however, our understanding of how CAMs contribute to the precise regulation of synaptic properties is still in its infancy.

Specificity of synaptic connections and synaptic properties at cellular and subcellular levels has been well documented. Several hypotheses have been proposed to account for synaptic specificity and the putative role of synaptic CAMs in shaping synaptic specificity. However, in only a few cases have synaptic CAMs been shown to instruct or participate in shaping synaptic specificity [4]. In the hippocampal CA1, synaptic specificity in excitatory circuits is mediated by the adhesion G protein-coupled receptors, latrophilin-2 (Lphn2), and Lphn3 [5]. This specificity was proposed to be primarily conferred by distinct expression patterns of the postsynaptic CAMs, fibronectin leucine rich transmembrane protein 2 (FLRT2) and teneurin-3 (Ten3), because Lphn2 and Lphn3 are capable of binding to both of these CAMs [1,5]. However, another study reported that heterophilic FLRT2/Ten3 and homophilic Ten3/Ten3 *trans*-interactions instruct the assembly of specific circuits encompassing the hippocampal CA1 and subiculum through a combination of reciprocal repulsions [6]. Moreover, Lphn2 and Lphn3 appear to redundantly shape synapse properties in the cerebellum [7]. Collectively, these observations suggest the hypothesis that synaptic properties are partly determined by inherently hardwired mechanisms in a region-specific manner and partly shaped by unique, context-dependent mechanisms. However, the generality of this concept has not been tested. Leucine-rich repeat (LRR)-containing proteins have emerged as key organizers of synaptic specificity, reflecting their cell-type–specific expression patterns [8,9]. Among other proteins, a subset of LRR proteins has been shown to participate in synaptic specificity. For example, netrin-G ligand 1 (NGL-1) and NGL-2 act similarly to Lphn2 and Lphn3 in shaping excitatory circuits in the hippocampal CA1 (S1 Table). In addition, a subset of LRR proteins exhibit modular, nonredundant functions in mediating synaptic specificity.

The Slit- and Trk-like (Slitrk) protein family consists of six members that perform paralog-specific synaptic functions [10–16]. Slitrk3 partners with neuroligin-2 (Nlgn2) to control GABAergic synapses, whereas Slitrk2 and Slitrk5 cooperate with tyrosine kinase receptor B (TrkB) to regulate glutamatergic synapses [13,15,17]. Whether these molecular interactions dictate synaptic specificity, however, has not been examined. A study employing a knockdown (KD)-based loss-of-function approach in rats demonstrated that Slitrk1 participates in regulating distinct excitatory synaptic properties in the excitatory circuits in the hippocampus [18]. However, this study did not examine whether other Slitrk paralogs similarly encode synaptic specificity or investigate the underlying mechanisms. Moreover,

PSD, postsynaptic density; PSD-95, post-synaptic density 95; qRT-PCR, quantitative reverse transcription-polymerase chain reaction; ROI, region of interest; SLM, *stratum lacunosum-moleculare*; SP, *stratum pyramidale*; SR, *stratum radiatum*; TEM, transmission electron microscopy; Ten3, teneurin-3; TrkB, tyrosine kinase receptor B; ΔCre, inactive Cre; WT, wild-type.

how Slitrk-associated neurological disorders are linked to specific impairment of Slitrk-mediated regulation of circuit properties remains unknown [17,19–21].

In the present study, we focused on two excitatory synapse-specific Slitrk paralogs, Slitrk1 and Slitrk2, asking the following key questions: (1) Are Slitrk1 and Slitrk2 involved in determining synaptic specificity in the hippocampal CA1? (2) If so, are the synapse-specific actions of Slitrk1 and Slitrk2 recapitulated in different types of hippocampal circuits (e.g., dentate gyrus [DG])? (3) What are the mechanisms underpinning the Slitrk-mediated synaptic specificity? (4) And is impaired Slitrk-mediated synaptic specificity linked to Slitrk dysfunction-associated neurological disorders? Using adult mice with conditional deletion of Slitrk1 or Slitrk2 in specified hippocampal microcircuits, in conjunction with extensive electrophysiological recordings of various synaptic properties and behavioral analysis, we identified a new principle of synaptic specificity. Specifically, we found that different paralogs in a single CAM family encode synaptic specificity in a context-dependent manner, likely through distinct transsynaptic and intracellular signaling pathways.

## Materials and methods

### Ethics statement

All procedures were performed in accordance with the Animal Protection Act of the Republic of Korea and the guidelines of the Institutional Animal Care and Use Committee (IACUC). The experimental protocols were approved by Daegu Gyeongbuk Institute of Science and Technology (DGIST) Administrative Panel on Laboratory Animal Care (Approval No. DGIST-IACUC-23112808-0004 and DGIST-IACUC-23063002-0000) and Yonsei University (Approval No. IACUC-202301-1617-01).

### Mice

Mice were given *ad libitum* access to food and water and were maintained in a controlled environment with a 12-hour light/dark cycle (lights on at 7:00 am).

**1. Slitrk1fl/f.** Slitrk1-cKO mice were generated by Cyagen. The mouse *Slitrk1* gene (GenBank accession number: NM_199065.2, Ensembl: Slitrk1 ENSMUSG00000075478) located on chromosome 14 contains a single exon, which was selected as the cKO region. The targeting vector was engineered by generating homology arms and cKO region by PCR with high-fidelity Taq, using BAC clones RP23-160B19 and RP23-438E14 from the C57BL/6J library as templates. Fragments were sequentially assembled into the targeting vector, consisting of a Neo cassette flanked by Frt sites and the cKO region flanked by loxP sites. Restriction enzyme digestion and sequencing were used to confirm the final targeting vector. C57BL/6 ES cells were used for gene targeting. After confirmation of correctly targeted ES clones by Southern blotting, clones were selected for blastocyst microinjection and subsequent chimera production. Heterozygous founders containing the conditional KO allele were confirmed as germline-transmitted by crossbreeding with *Flp*-deleter mice.

**2. *Slitrk2-V89Mf/f*.**  The *Slitrk2* V89M conditional knock-in (cKI) mouse was created by Beijing Biocytogen using the homologous recombination method. In brief, one loxP site was inserted upstream of exon 2 and downstream of the 3′ UTR, and the p.V89M mutation was introduced into exon 3 using an overlap extension PCR method. The 3XSV40 polyA STOP and FRT-flanked Neo resistance-positive selection cassette was inserted downstream of 3′ UTR. The fidelity of targeting vector construction was verified by full sequencing. After linearization, the targeting vector was transfected into C57BL/6J embryonic stem (ES) cells by electroporation. Five positive ES clones were identified by Southern blot analysis and karyotype analysis. Positive ES clones were injected into Balb/c blastocysts and implanted into pseudopregnant females. Chimeric male mice were then crossed with FLP females to obtain F1 mice carrying the recombined allele containing the loxP elements, with the Neo selection cassette removed. F1 mice were validated for germ line transmission of the recombination event using a PCR genotyping strategy. Elimination of the cassette in offspring was analyzed by PCR using the primers, 5′-GCCTCTGAGACATGTCAAGCACTG-3′ (Frt-F) and 5′-GAATTTGAGAGGTTTTGCAGGCTCC-3′ (Frt-R). Male and female heterozygous mice were crossed to produce homozygous mutant mice. The *Slitrk2*-floxed [17] and *Ntrk2*-floxed lines were described previously [22].

## Cell culture

Human Embryonic Kidney 293T (HEK293T; ATCC; Cat #CRL-3216) cells were cultured in Dulbecco's Modified Eagle's Medium (DMEM; Welgene) supplemented with 10% fetal bovine serum (FBS; Tissue Culture Biologicals) and 1% penicillin-streptomycin (Thermo Fisher) at 37 °C in a humidified 5% $CO_2$ atmosphere.

## Expression vectors

pDisplay-SLITRK missense variants (SLITRK1 N162R, SLITRK2 V89M, and SLITRK2 N166R) were created by PCR-based mutagenesis with PrimeSTAR HS DNA polymerase (Takara) using pDisplay-SLITRK1 WT or pDisplay-SLITRK2 WT as a template. pAAV-SLITRK variants (pAAV-SLITRK1 WT-T2A-GFP, pAAV-SLITRK1 N162R-T2A-GFP, pAAV-SLITRK2 WT-T2A-GFP, pAAV-SLITRK2 V89M-T2A-GFP, and pAAV-SLITRK2 N166R-T2A-GFP) were generated by PCR-amplifying and subcloning the full-length sequence of the respective human *SLITRK* into the pAAV-T2A-GFP vector at *Xba*I and *Bam*HI sites. pDisplay-SLITRK1 WT, pDisplay-SLITRK2 WT, and pCMV-SPORT6-TrkB as well as pCMV-IgC and pCMV-IgC PTPσ$^{A+B+}$ constructs were previously described [17,23]. The following constructs were purchased from Addgene: pAAV-EF1α-DIO-hM4D(Gi)-mCherry (Cat #50461) and pAAV-hSyn-HA-hM3D(Gq)-IRES-mCitrine (Cat #50475).

## Antibodies

The following antibodies were obtained from commercial sources: goat polyclonal anti-Slitrk1 (R&D Systems; Cat #AF3009-SP; RRID: AB_2190304); rabbit polyclonal anti-Slitrk2 (Invitrogen; Cat #PA5-20471; RRID: AB_11156827); guinea pig polyclonal anti-VGLUT1 (Synaptic Systems; Cat #135 304; RRID: AB_887878); mouse monoclonal anti-PSD-95 (clone K28/43; BioLegend; Cat #MMS-5182; RRID: AB_2564750); mouse polyclonal anti-gephyrin (Synaptic Systems; Cat #147 011; RRID: AB_887717); rabbit monoclonal anti-TrkB (clone 80E3; Cell Signaling; Cat #4603; RRID: AB_2155125); rabbit polyclonal anti-GluN1 (Millipore; Cat #MAB363; RRID: AB_94946); rabbit polyclonal anti-GABA$_A$Rγ2 (Synaptic Systems; Cat #224 003; RRID: AB_2263066); guinea pig polyclonal anti-SHANK2 (Synaptic Systems; Cat #162 204; RRID: AB_2619861); guinea pig polyclonal anti-VGLUT1 (Millipore; Cat #AB5905; RRID: AB_2301751); mouse polyclonal anti-HA (BioLegend; Cat #901501; RRID: AB_2565006); mouse polyclonal anti-NeuN (Millipore; Cat #MAB377; RRID: AB_2298772); mouse monoclonal anti-β-actin (Santa Cruz; Cat #sc-47778; RRID: AB_2714189); goat polyclonal anti-GFP (Rockland; Cat #600-101-215; RRID: AB_218182); donkey anti-goat IgG Alexa Fluor 488 (Abcam; Cat #ab150129; RRID: AB_2687506); goat anti-rabbit IgG Alexa Fluor 488 (Abcam; Cat #ab150077; RRID: AB_2630356); donkey anti-guinea pig Alexa Fluor 594 (Jackson ImmunoResearch; Cat # 706-585-145; RRID:

AB_2340474); donkey anti-mouse IgG Alexa Fluor 647 (Abcam; Cat #ab150111; RRID: AB_2890625); Cy3-AffiniPure donkey anti-rabbit IgG (Jackson ImmunoResearch; Cat #711-165-152; RRID: AB_2307443); and Cy3-donkey anti-guinea pig IgG (Jackson ImmunoResearch; Cat #706-035-148; RRID: AB_ 2340447). The following in-house–prepared antibodies were previously described: rabbit polyclonal anti-Slitrk2 (JK177; RRID: AB_2892626) [17]; rabbit polyclonal anti-pan-SHANK (1172; RRID: AB_2810261) [24]; rabbit polyclonal anti-PSD-95 (JK016; RRID: AB_2722693) [24]; and rabbit polyclonal anti-GluA1 (1193; RRID: AB_2722772; a gift from Dr. Eunjoon Kim [KAIST/IBS, Korea]) [25].

### Quantitative reverse transcription-polymerase chain reaction (qRT-PCR)

Hippocampal tissues of *Slitrk1*-cKO mice were used for mRNA quantification. Mouse brain tissue samples were each lysed in 250 µl of TRIzol (Invitrogen) according to the manufacturer's protocol; following phenol-chloroform extraction, RNA was precipitated by isopropanol. cDNA was synthesized from total RNA using a ReverTra Ace-a kit, as instructed by the manufacturer (Takara Bio). Quantitative reverse PCR was performed on a CFX96 Touch Real-Time PCR system (BioRad) using 1 µL of cDNA. The following targets were amplified by qPCR using the indicated primer pairs: *Slitrk1*, 5′-CACCCTACCTGCTAATGTATTCC-3′ (forward) and 5′-TGCTCCAAGACCTCCTCATA-3′ (reverse); *Slitrk2*, 5′-GCAG AGCTTGCAGTATCTCTATT-3′ (forward) and 5′-GGACCTCAGCAGGTTGTTATT-3′ (reverse); *Slitrk3*, 5′-TGAGAAGAAT GGTGGGGTGGTG-3′ (forward) and 5′-TTTGGGGGTTTGGTGGTCTGTG-3′ (reverse); *Slitrk4*, 5′-GTGG CCTGAAAGAGTCAGAAA-3′ (forward) and 5′-TCCTGGTATCCACATGCAATAAA-3′ (reverse); and *Slitrk5*, 5′-GTGGA CGAAGTGATCTGTAAGG-3′ (forward) and 5′-GGAAACCACCACATCAGAGTAG-3′ (reverse).

### Staining for surface/intracellular protein levels

HEK293T cells were transfected with the indicated SLITRK expression vectors. Forty-eight hours after transfections, cells were washed twice with phosphate-buffered saline (PBS), fixed by incubating with 3.7% formaldehyde for 10 min at 4 °C, and blocked with 3% horse serum/0.1% bovine serum albumin (BSA; crystalline grade) in PBS for 15 min at room temperature. Surface-expressed SLITRK protein was then detected by staining with mouse anti-HA antibody at room temperature. After 90 min, cells were washed twice with PBS and incubated with fluorescein isothiocyanate (FITC)-conjugated anti-mouse antibodies for 1 hour at room temperature. Cells were then permeabilized by incubating with PBS containing 0.2% Triton X-100 for 10 min at 4 °C, followed by incubation with rabbit anti-HA antibody for 90 min at room temperature to label intracellularly expressed SLITRK proteins. Permeabilized cells were then incubated with Cy3-conjugated anti-rabbit secondary antibodies.

### Cell-surface binding assays

The Ig-fusion protein IgPTPσ$^{A-B+}$ and IgC control were produced in HEK293T cells. Cells were transfected with the indicated IgC plasmids and 48 hours later soluble Ig-fused proteins were purified using protein A-Sepharose (GE Healthcare) as previously described [26]. Ig-fused proteins were eluted with 0.1 M glycine (pH 2.5) and immediately neutralized with 1 M Tris·HCl (pH 8.0). HEK293T cells expressing HA-tagged SLITRK variants (SLITRK1 WT, SLITRK1 N162R, SLITRK2 WT, SLITRK2 N166R, SLITRK2 ΔISQL, or SLITRK2 V89M) were incubated with 10 µg/ml of the indicated Ig-fused proteins for 2 hours at 37 °C with gentle agitation. The immunocytochemistry procedures used were as described in the previous section. Images were acquired using a confocal microscope (LSM700; Zeiss).

### Coimmunoprecipitation assays in heterologous cells

HEK293T cells were transfected with pCMV-SPORT6-TrkB and the indicated pDisplay-SLITRK2 plasmids. After 48 hours, the transfected HEK293T cells were rinsed with ice-cold PBS and solubilized in lysis buffer (20 mM Tris pH 7.4, 1.0% Triton X-100, 0.1% SDS, 150 mM NaCl, 10% glycerol, 0.2 mM PMSF, 1 µg/ml aprotinin, 1 µg/ml leupeptin, 1 µg/ml pepstatin,

1 mM $Na_3VO_4$). After centrifugation at 18,000 × $g$, the supernatants were incubated with 1 μg of the appropriate antibody overnight at 4 °C. Thereafter, 30 μl of a 1:1 suspension of protein A-Sepharose (Incospharm Corporation) was added, and the mixture was incubated for 2 hours at 4 °C with gentle rotation. Immune complexes were then resolved by SDS-PAGE and immunoblotted with the indicated antibodies. Coimmunoprecipitation experiments were repeated at least three times, and quantified results are expressed as the amount of protein co-precipitated relative to the input amount. Immunoblot images presented in figures are representative of results from multiple biological replicates.

### Primary neuronal culture, transfection, immunocytochemistry, and image acquisition and analysis

Rat hippocampal cultured neurons were prepared from E18 rat brains, cultured on coverslips coated with poly-D-lysine, and grown in neurobasal medium supplemented with B-27 (Thermo Fisher Scientific), 0.5% FBS, 0.5 mM GlutaMax (Thermo Fisher Scientific), and sodium pyruvate (Thermo Fisher Scientific), as previously described [27]. Neurons were co-transfected at DIV8 with vectors encoding L-315 [28] and the indicated Slitrk2 variants using a CalPhos Transfection Kit (Takara), and subsequently immunostained at DIV14. For immunocytochemistry, cultured neurons were fixed with 4% paraformaldehyde/4% sucrose, permeabilized with 0.2% Triton X-100 in PBS, immunostained with the indicated primary antibodies, and detected with the indicated Cy3- or FITC-conjugated secondary antibodies (Jackson ImmunoResearch). Transfected neurons were chosen randomly and images were acquired using a confocal microscope (LSM800; Zeiss) with a 63 × objective lens; all image settings were kept constant. Z-stack images were converted to maximal intensity projections and analyzed to obtain the size, intensity, and density of puncta immunoreactivities derived from marker proteins. Quantification was performed in a blinded manner using the MetaMorph software (Molecular Devices).

### Semi-quantitative immunoblotting

Hippocampal tissues from the indicated *Slitrk* transgenic mice were lysed in lysis buffer (150 mM NaCl, 50 mM Tris, 1% Triton X-100 (pH 7.4)). The homogenized tissue was centrifuged at 12,000 × $g$ for 15 min, and then the supernatants were collected and used for immunoblotting. The following antibodies were used: human polyclonal anti-Slitrk1 (0.5 μg/ml); rabbit polyclonal anti-Slitrk2 (JK177; 1 μg/ml); polyclonal anti-GluN1 (1 μg/ml); polyclonal anti-GluA1 (1193; 1:1,000); mouse polyclonal anti-gephyrin (1 μg/ml); rabbit polyclonal anti-PSD-95 (JK016; 1:1,000); rabbit polyclonal anti-GABA$_A$Rγ2 (1 μg/ml); rabbit monoclonal anti-TrkB (1 μg/ml); mouse monoclonal anti-β-actin (1 μg/ml); and mouse polyclonal anti-HA (1 μg/ml).

### Drugs and chemicals

The following drugs and chemicals were purchased from Sigma-Aldrich: NaCl (Cat #S9888), KCl (Cat #P3911), $KH_2PO_4$ (Cat #60218), $NaHCO_3$ (Cat #S8761), D-glucose (Cat #G8644), sucrose (Cat #S9378), $SrCl_2$ (Cat #439665), $CaCl_2$ (Cat #223506), $MgCl_2$ (Cat #M8266), CsCl (Cat #289329), ethylene glycol-bis (2-aminoehylether)-N,N,N',N-tetraacetic acid (Cat #E4378), adenosine 5′-triphosphatate magnesium salt (Cat #A9187), guanosine 5′triphosphatate sodium salt hydrate (Cat #G8877), cesium methanesulfonate (Cat #C1426), tetraethylammonium chloride (Cat #T2265), phosphocreatine disodium salt hydrate (Cat #P7936), potassium gluconate (Cat #P1847), CNQX (Cat #C127), DNase I (Cat #D5307), 10 x Reaction buffer (Cat #R6273), and stop solution (Cat #S4809). The following agents were purchased from Tocris: picrotoxin (Cat #1128), D-AP5 (Cat #0106), and QX-314 bromide (Cat #1014). Deschloroclozapine (DCZ) was purchased from MedChemExpress (Cat #HY-42110). Proteinase K was purchased from Millipore (Cat #70663).

### Production of recombinant viruses

   **1. Lentiviruses.** Lentiviruses were produced by transfecting HEK293T cells with the indicated lentiviral vectors, together with psPAX2 and pMD2.G at a 2:2:1 ratio. Lentiviruses were harvested 72 hours after transfections by collecting

the media from transfected HEK293T cells and centrifuging at 1,000 × *g* to remove cellular debris, as previously described [26].

2. **Adeno-associated viruses (AAVs).** For high-efficiency transfections, AAVs were packaged with pHelper and AAV1.0 (serotype 2/9) capsids, as previously described [29,30]. Briefly, HEK293T cells were co-transfected with pHelper and pAAV1.0$_{2/9}$ (serotype 2/9) vectors together with the indicated pAAV vector. Cells were harvested 72 hours later, lysed, mixed with 40% polyethylene glycol and 2.5 M NaCl, and centrifuged at 2,000 × *g* for 30 min. The resulting pellets were resuspended in HEPES buffer (20 mM HEPES, 115 mM NaCl, 1.2 mM $CaCl_2$, 1.2 mM $MgCl_2$, 2.4 mM $KH_2PO_4$), mixed with an equal volume of chloroform, and centrifuged at 400 × *g* for 5 min. The supernatants were concentrated three times with a Centriprep centrifugal filter (15 ml, 4,310; Millipore) at 1,220 × *g* for 5 min each and then with an Amicon Ultra centrifugal filter (0.5 ml, 3 K MWCO; Millipore) at 16,000 × *g* for 10 min. The infectious titer of viruses was assessed by qRT-PCR detection of EGFP sequences, with subsequent reference to a standard curve generated using the pAAV-U6-EGFP plasmid.

## Stereotactic injections

Stereotaxic injections were performed using *Slitrk1*$^{f/f}$, *Slitrk2*$^{f/f}$, *Slitrk1/2*$^{f/f}$, *Slitrk2*-V89M$^{f/f}$, and *Ntrk2*$^{f/f}$ mice. Mice were anesthetized by inhalation of isoflurane. The CA1 and DG, respectively, were targeted using the following coordinates (relative to Bregma): AP: −2.5, ML: 1.5, DV: −1.3 (CA1) and AP: −2.5, ML: 1.5, DV: −1.7 (DG). Both brain hemispheres were injected with 350 nl of virus (titer ≥ 1 × $10^{10}$ genomes/μl) at a rate of 0.1 μl/min using a NanoFil syringe and a Nanoliter 2010 Injector (World Precision Instruments).

## Chemogenetic manipulation with DREADDs

Male *Slitrk*-floxed mice (5–6 weeks old) were anesthetized and injected with 300 nl of AAV-Cre+DIO-hM4D(Gi)-mCherry or AAV-DIO-hM3D(Gq)-mCherry in the CA1 region. Two weeks after the injections, slices were prepared for electrophysiological recordings. A solution containing DCZ or vehicle (saline) was bath-applied to the aCSF 1 hour prior to the electrophysiological experiments. DCZ was dissolved in 1–2% dimethyl sulfoxide (DMSO) in saline to a final concentration of 125 nM.

## Epitope-preserving magnified analysis of proteome (eMAP)

Six- to eight-week-old WT male C57BL/6 mice (Ontario), *Slitrk1*-cKO, or *Slitrk2*-cKO mice were used for eMAP analyses, which were performed as previously described [31] with minor modifications. Mice were anesthetized by isoflurane and euthanized via transcardial perfusion with 20 mL of ice-cold 1 × PBS. The mice then were perfused with 20 mL of ice-cold fixative solution (4% (w/v) paraformaldehyde (PFA) in PBS at 4 °C). The brains were obtained and incubated in an ice-cold fixative solution overnight at 4 °C, followed by incubation at room temperature for 3 hours. After post-fixation, the brains were washed overnight in 1 × PBS at room temperature. Coronal sections of the brains were prepared at a thickness of 160 μm using a vibratome (VT1200S; Leica). For an accurate comparison of staining conditions, brain sections from 1.8 to 2.2 mm posterior of the bregma were sampled for analysis of the hippocampal region. The eMAP process was performed as previously described [32]. Briefly, the PFA-fixed tissues were incubated at 4 °C in an eMAP solution (30% (w/v) acrylamide, 10% (w/v) sodium acrylate, 0.1% (w/v) N,N′-methylenebis(acrylamide), 0.03% (w/v) VA-044 in PBS) and mounted between two slide glasses with extra gel solution. The slides were secured using Blu-Tack adhesive to create a gelling chamber, which was incubated for 3.5 hours at 37 °C under nitrogen purging (10 psi). The resulting hydrogel-embedded tissues were incubated overnight in a denaturation buffer (300 mM SDS, 100 mM sodium sulfite, and 10 mM boric acid, pH 9) at 37 °C. The samples were subsequently incubated in preheated denaturation solution at 95 °C for 10 min, immediately transferred to PBST (0.1% (v/v) Triton X-100 in PBS), and washed twice in PBST for 3 hours at 37 °C. For immunostaining of eMAP-processed tissues, the samples were incubated overnight at 37 °C in primary antibody

solution (1: 200 dilution in 5% (v/v) normal donkey serum (NDS) in PBST), washed three times with PBST at room temperature for 30 min each, incubated overnight at 37 °C in the indicated secondary antibody solution (1:200 dilution in 5% (v/v) NDS in PBST), and washed three times with PBST at room temperature for 30 min each. To analyze the expression patterns of Slitrk proteins in the eMAP tissues, we performed immunostaining as described above. Fluorescent images were captured using a white light laser confocal microscope (TCS SP8 X; Leica) with a 63 ×/1.30 NA glycerol immersion objective. Imaging was performed in the SR and SLM regions of the CA1 hippocampus and the OML and MML regions of the DG. At each location, 5-μm (physical unit) Z-stacks were collected with a Z-plane separation distance of 0.33 μm. To validate the activity of our Slitrk1 and Slitrk2 antibodies and evaluate the effects of conditional knockouts, we analyzed image stacks for the total number of synaptic puncta and the colocalization of Slitrks with the postsynaptic markers, PSD-95 and SHANK2 using the DiAna plugin of Fiji [33]. For each fluorescent channel, thresholding was performed manually to select only the signal in the focal plane of each stack; the same threshold values were used for images from the same tissue. The 3D colocalization percent was calculated by dividing the number of colocalized Slitrk signals by the total number of thresholded Slitrk signals. For each mouse ($n = 3$ mice/group) three separate gels were analyzed and the measurements were averaged for each mouse. All data are presented as means ± SEMs ($n = 3$ mice/group).

## Immunohistochemistry, imaging, and analyses

Adult mice (WT, *Slitrk1*-cKO or *Slitrk2*-cKO) were anesthetized by inhalation of isoflurane and transcardially perfused with 4% PFA. After overnight post-fixing in 4% paraformaldehyde, the brain was sliced at a thickness of 40 μm using a Vibratome (VT1200S; Leica Biosystems). Sections were dried on Superfrost Plus Adhesion Microscope Slides (Epredia) and then sprayed with 2% Triton X-100 in PBS containing 5% bovine serum albumen (PBS-BH) and 5% horse serum. The tissue was sprayed with antibody, diluted in PBS-BH, and then left at 4 °C overnight. The tissue was then washed twice with 1×PBS, after which a secondary antibody staining, diluted in PBS-BH, was sprayed on the tissue and incubated at room temperature for 2 hours. Tissues were washed twice with 1× PBS and then mounted using a mounting solution containing 4′,6-diamidino-2-phenylindole (DAPI; VECTASHELD; H-1200). Tile scans were acquired by confocal microscopy (LSM700; Zeiss). Intensity quantifications were performed in MetaMorph by measuring a plot profile of the selected hippocampal region (a consistent region of interest (ROI) was used for all images), averaging the intensity values across the same distance point for all images, normalizing to the minimum and maximum values, and binning the data every 10% of depth from the CA1 SO layer. Fluorescence images were processed using MetaMorph software (Molecular Devices). A Raw Pass filter was applied with both X and Y values set to 2 to reduce background noise and preserve image detail. Original images were acquired as 1,024 × 1,024 pixels, 24-bit RGB TIFF files (8 bits per channel) at a resolution of 3.41 × 3.41 inches. To isolate fluorescent signals from the background, color thresholding was applied in the HSB color space using the Threshold Color tool. Hue, Saturation, and Brightness parameters were adjusted to selectively extract high-intensity red-fluorescent pixels while minimizing background noise. Specifically, the threshold ranges were set as follows: Hue = 35–50, Saturation = 0–255, and Brightness = 100–255. The 'Threshold color' was set to "Red," and the 'Dark background option' was unchecked. These parameters were optimized to detect bright red puncta while excluding dim or nonspecific signals. Following thresholding, the images were converted to binary masks for quantitative analysis. All images were processed under identical threshold conditions to ensure consistency across samples.

## Tissue processing and image acquisition for transmission electron microscopy

Mice were anesthetized by inhalation of isoflurane and transcardially perfused with 2% paraformaldehyde and 2.5% glutaraldehyde in 0.15 M cacodylate buffer (pH 7.4) containing 2 mM $CaCl_2$ and 0.2% dextrose. The extracted brains were stored overnight at 4 °C in fresh fixative solution and sliced into 150-μm-thick coronal sections in cold 0.15 M cacodylate buffer using a vibratome (Leica VT1000S; Leica Microsystems, Vienna, Austria). Slices were dissected into DG

and CA1 regions under a microscope, then treated with a cacodylate buffer containing 2% osmium tetroxide ($OsO_4$) and 1.5% potassium ferrocyanide for 1 hour on ice, followed by staining *en bloc* with 1% uranyl acetate overnight at 4 °C. The tissues were dehydrated using a graded series of ethanol and then incubated in ice-cold dry acetone for 10 min. The samples were gradually equilibrated with Epon 812 resin (EMS, Hatfield, PA; Cat #14120) by placing them in a mixture of resin and acetone. The tissue was placed in embedding tubes with fresh Epon mixture and cured at 60 °C in a dry oven for 2 days. Each tissue block was trimmed flat using a glass knife, cut into 70 nm sections using an ultramicrotome (EM UC7, Leica Microsystems) with a diamond knife (Diatome), and collected on formvar-coated single-slot grids. The ultrathin sections on the grid were post-stained with lead citrate (EMS, Hatfield, PA; Cat #22410) for 1 min to enhance contrast. Images were captured on a Tecnai G2 TEM (Field Electron and Ion Company) operating at 120 kV and equipped with an sCMOS camera. Synapses were imaged at high magnification (14,000 ×) in the SR layer and SLM layer of the CA1, which are innervated by the Schaeffer collaterals and temporo-ammonic pathways. Synapses were also imaged in the MML and OML of the DG, which are innervated by the medial perforant pathway and lateral perforant pathway, respectively.

## Electron microscopy image analyses

All features in an individual synapse were quantified using ImageJ (National Institutes of Health). The following subcellular structures were measured: presynaptic bouton, postsynaptic spine, active zone (AZ), postsynaptic density (PSD), and vesicles. Synapses with unclear PSD or AZ were excluded from the analysis. Custom macro scripts in Fiji were used to measure two-dimensional data, including the area, length, and number of each synaptic structure in individual images. The areas of the bouton and spine were quantified by obtaining the area value for each synaptic structure. The length of the PSD and AZ was measured by summing the electron-dense length across the synapse. Vesicles that were located within 25 nm of the AZ were designated as docked vesicles. The vesicle density was calculated by normalizing the number of vesicles to the AZ length for each synapse. A distance map was created for the AZ in the image, and the Euclidean distance from the AZ to the center of the vesicle was calculated. The relative frequency at each location was obtained by expressing the distance of vesicle from the AZ in 25-nm bins.

## Electrophysiology

Hippocampal slices (300 μm) were prepared from male mice aged 6–8 weeks. Following anesthesia with isoflurane, mice were euthanized and their brains were swiftly extracted and placed in a chilled, oxygenated (95% $O_2$ and 5% $CO_2$) solution containing reduced $Ca^{2+}$ and elevated $Mg^{2+}$ levels, with the composition 3.3 mM KCl, 1.3 mM $NaH_2PO_4$, 26 mM $NaHCO_3$, 11 mM D-glucose, 0.5 mM $CaCl_2$, 10 mM $MgCl_2$, and 211 mM sucrose. Hippocampal slices were prepared using a vibratome (VT1000s; Leica) and transferred to a storage chamber filled with oxygenated artificial cerebrospinal fluid (aCSF) containing 124 mM NaCl, 3.3 mM KCl, 1.3 mM $NaH_2PO_4$, 26 mM $NaHCO_3$, 11 mM D-glucose, 2 mM $CaCl_2$, and 1 mM $MgCl_2$. Slices were incubated at 30 °C for at least 60 min prior to experimentation. Slices were then transferred into the recording chamber and constantly perfused with standard aCSF oxygenated with a mixture of 95% $O_2$ and 5% $CO_2$. All experiments were performed at 27–30 °C, and slices were used within 4 hours. Only cells with access resistance ($R_a$) < 30 MΩ were analyzed. Cells were discarded if their resting membrane potential was > −45 mV. All recordings were obtained using a Multiclamp 700B amplifier and DigiData 1550B Digitizer. We noted that baseline electrophysiological values (e.g., mEPSC frequency) for control groups exhibited some variability between independent experiments. This degree of variation is commonly observed in the field for such complex recordings and, in our study, may also have been influenced by the necessary use of different AAV-ΔCre-injected transgenic lines as respective controls for different conditional knockout experiments. Therefore, to ensure valid comparisons and account for this inter-experiment variability, all data within a given experimental set were first normalized to the mean of their respective internal control group before statistical comparisons were made across conditions.

**1. CA1 pyramidal neuron recordings.** Spontaneous excitatory postsynaptic currents (sEPSCs) were assessed by whole-cell recordings from hippocampal CA1 pyramidal neurons using glass pipettes (3–8 MΩ) filled with a solution containing 145 mM CsCl, 5 mM NaCl, 10 mM HEPES, 10 mM EGTA, 4 mM Mg-ATP, and 0.3 mM Na-GTP, adjusted to pH 7.2–7.3 with CsOH. Cellular voltage was clamped at −70 mV. sEPSCs were isolated by inhibiting GABA$_A$ receptors using external application of 50 μM picrotoxin. To isolate miniature excitatory postsynaptic currents (mEPSCs), 1 μM TTX is additionally added to external solution. For measurement of evoked synchronous excitatory postsynaptic currents (eEPSCs) and evoked asynchronous excitatory postsynaptic currents (aEPSCs), patch pipettes were filled with an internal solution consisting of 130 mM Cs-methanesulfonate, 5 mM TEA-Cl, 8 mM NaCl, 0.5 mM EGTA, 10 mM HEPES, 4 mM Mg-ATP, 0.4 mM Na-GTP, 1 mM QX-314, and 10 mM disodium phosphocreatine, adjusted to pH 7.2–7.3 with CsOH. Cells were voltage clamped at −70 mV (for AMPAR-EPSCs) or +40 mV (for NMDAR-EPSCs; 50 ms after stimulation). Electrical stimulation was applied using a concentric bipolar electrode (FHC), placed on the SR layer in the hippocampal CA1 (to record eEPSCs and aEPSCs at SC synapses) or on the SLM layer (to record eEPSCs and aEPSCs at TA synapses). eEPSCs and aEPSCs were isolated by inhibiting GABA$_A$ receptors using the external application of 50 μM picrotoxin. For eEPSC recording from CA1 pyramidal neurons, slices were perfused with oxygenated aCSF, containing 124 mM NaCl, 3.3 mM KCl, 1.3 mM NaH$_2$PO$_4$, 26 mM NaHCO$_3$, 11 mM D-glucose, 3 mM CaCl$_2$, and 3 mM MgCl$_2$. eEPSC input-output (eEPSC I-O) was recorded by applying electrical stimulation ranging from 20 to 100 μA in 20-μA increments. Average eEPSCs I-O values were measured from three consecutive sweeps. Paired-pulse ratios (PPRs) were assessed by administering pairs of stimuli at a frequency of 20 Hz. PPRs were calculated as the ratio, 2nd eEPSC/1st eEPSC. Average PPRs were analyzed from three consecutive sweeps. Recording of aEPSCs was conducted in aCSF, with Ca$^{2+}$ replaced by Sr$^{2+}$ (124 mM NaCl, 3.3 mM KCl, 1.3 mM NaH$_2$PO$_4$, 26 mM NaHCO$_3$, 11 mM D-glucose, 8 mM SrCl$_2$, and 3 mM MgCl$_2$). Using 8 mM SrCl$_2$ led to initial modest synchronous responses, followed by an interval of heightened quantal release at both SC synapses and TA synapses. aEPSCs were induced by applying paired stimulation to the SR or SLM layer, delivered as 10 consecutive sweeps at a frequency of 10 Hz [34,35]. The intrinsic excitability of CA1 pyramidal neurons was measured using glass pipettes filled with an intracellular solution consisting of 130 mM K-gluconate, 20 mM KCl, 0.2 mM EGTA, 10 mM HEPES, 4 mM Mg-ATP, 0.3 mM Na-GTP, and 10 mM disodium phosphocreatine, adjusted to pH 7.2–7.3 with KOH. Current was injected into CA1 pyramidal neurons over a range of 0–300 pA at 25-pA increments.

**2. DG granule neuron recordings.** sEPSCs were recorded using patch pipettes (4–8 MΩ) filled with a solution containing 145 mM CsCl, 5 mM NaCl, 10 mM HEPES, 10 mM EGTA, 4 mM Mg-ATP, and 0.3 mM Na-GTP, adjusted to pH 7.2–7.3 with CsOH. Cellular voltage was clamped at −70 mV. sEPSCs were isolated by inhibiting GABA$_A$ receptors using the external application of 50 μM picrotoxin. eEPSCs and aEPSCs were assessed using patch pipettes filled with an internal solution consisting of 130 mM Cs-methanesulfonate, 5 mM TEA-Cl, 8 mM NaCl, 0.5 mM EGTA, 10 mM HEPES, 4 mM Mg-ATP, 0.4 mM Na-GTP, 1 mM QX-314, and 10 mM disodium phosphocreatine, adjusted to pH 7.2–7.3 with CsOH. Cells were voltage clamped at −70 mV (for AMPAR-EPSCs) or +40 mV (for NMDAR-EPSCs; 50 ms after stimulation). For selective activation of MPP or LPP afferents in the DG, monopolar stimulation electrodes were positioned based on the established laminar organization of these specific pathways. We verified input identity based on short-term plasticity signatures: MPP stimulation induced paired-pulse depression, whereas LPP stimulation elicited paired-pulse facilitation, as previously described [36]. Electrical stimulation was applied using a concentric bipolar electrode, placed on the MML in the DG (to record eEPSCs and aEPSCs at MPP synapses) or on the OML (to record eEPSCs and aEPSCs at LPP synapses). eEPSCs and aEPSCs were isolated by inhibiting GABA$_A$ receptors using the external application of 50 μM picrotoxin. For recording of sEPSCs and eEPSCs from DG granule neurons, slices were perfused with oxygenated aCSF, containing 124 mM NaCl, 3.3 mM KCl, 1.3 mM NaH$_2$PO$_4$, 26 mM NaHCO$_3$, 11 mM D-glucose, 2 mM CaCl$_2$, and 1 mM MgCl$_2$. aEPSC recordings were carried out in aCSF with Ca$^{2+}$ replaced by Sr$^{2+}$ (124 mM NaCl, 3.3 mM KCl, 1.3 mM NaH$_2$PO$_4$, 26 mM NaHCO$_3$, 11 mM D-glucose, 8 mM SrCl$_2$, and 3 mM MgCl$_2$). eEPSC I-O was recorded by applying electrical stimulation over a range of 20–100 μA at 20-μA increments. Average eEPSC I–O was measured from three consecutive

sweeps. For assessing PPRs, pairs of stimuli were administered at a frequency of 20 Hz. PPRs were calculated as the ratio, 2nd eEPSC/1st eEPSC. Average PPRs were analyzed from three consecutive sweeps. aEPSCs were induced by administering paired stimulations to the MML or OML, delivered as 10 consecutive sweeps at a frequency of 20 Hz. The intrinsic excitability of DG granule neurons was assessed using glass pipettes filled with an intracellular solution consisting of 130 mM K-gluconate, 20 mM KCl, 0.2 mM EGTA, 10 mM HEPES, 4 mM Mg-ATP, 0.3 mM Na-GTP, and 10 mM disodium phosphocreatine, adjusted to pH 7.2–7.3 with KOH. The current was injected into DG granule neurons over a range of 0–300 pA in 25-pA increments.

### Barnes maze mouse behavioral analyses

The Barnes labyrinth test was administered as previously described [17]. In brief, a 95-cm diameter, white circular platform with 20 holes evenly distributed around the perimeter was prepared. The hole above the escape box was defined as the target. Bias based on olfactory or proximal cues within the labyrinth was avoided by rotating the maze daily, with the spatial placement of the target remaining constant with distal visual room cues. Starting on day 1, three trials were conducted per day for 7 consecutive days. On day 10, the first probe trial was performed without the escape box to confirm that the spatial task was acquired by distal environmental room cue navigation. The second probe trial was conducted on day 24. During the acquisition and probing trials, velocity, error score, total distance, initial visit time to escape box, and total latency were measured with a top-view infrared camera and analyzed using EthoVision XT 15 (Noldus) behavioral tracking software.

### Statistical analyses

Data analyses and statistical tests were performed using GraphPad Prism 7.0 software (RRID: SCR_002798). All data are expressed as means ± standard errors unless stated otherwise. All experiments were performed using at least three separate mice or independent cultures. No statistical methods were used to predetermine sample size; experiments were not randomized; and investigators were not blinded to allocation during experiments and outcome assessment. Data were compared using Student *t* test or one-way analysis of variance (ANOVA) using a nonparametric Kruskal–Wallis test, followed by Dunn's multiple comparison test for *post hoc* group comparisons, *t* test or Mann–Whitney *U* test; '*n*' numbers used are presented in figure legends. Numbers shown indicate replicates, and tests used to determine statistical significance are stated in the text and legends of figures depicting the results of the respective experiments. A *p*-value < 0.05 was considered statistically significant, and individual *p*-values are indicated in the respective figure legend.

## Results

### Expression profiles of Slitrk1 and Slitrk2 proteins in the adult mouse hippocampus

Previous studies have shown that all Slitrks exhibit a similar ability to induce presynaptic assembly in heterologous synapse-formation assays [10,11]. However, in hippocampal cultured neurons and CA1 pyramidal neurons, Slitrk1 and Slitrk2 specifically function at excitatory synapses, whereas Slitrk3 exerts its synapse-promoting actions specifically at GABAergic synapses [10,11,13,14,17,37]. Intriguingly, Slitrk1 exhibits a laminar distribution in the hippocampal CA1 with higher expression in the *stratum lacunosum-moleculare* (SLM) layer compared to the *stratum radiatum* (SR) layer [18]. However, these previous studies did not investigate the distribution pattern of Slitrk2 in the hippocampus. Thus, we performed immunohistochemistry in adult mouse brain sections using a new commercial anti-Slitrk2 antibody, together with commercially available anti-Slitrk1 antibodies; both commercial antibodies were further validated using the corresponding *Slitrk*-cKO mice (see below for details; S1A–S1D Fig). These immunofluorescence analyses revealed that the expression of Slitrk1 and Slitrk2 proteins is distributed across hippocampal CA1 and DG subfields (Fig 1A–1I), partly consistent with previous results [18]. However, Slitrk1 and Slitrk2 proteins were observed to be enriched to different extents in specific

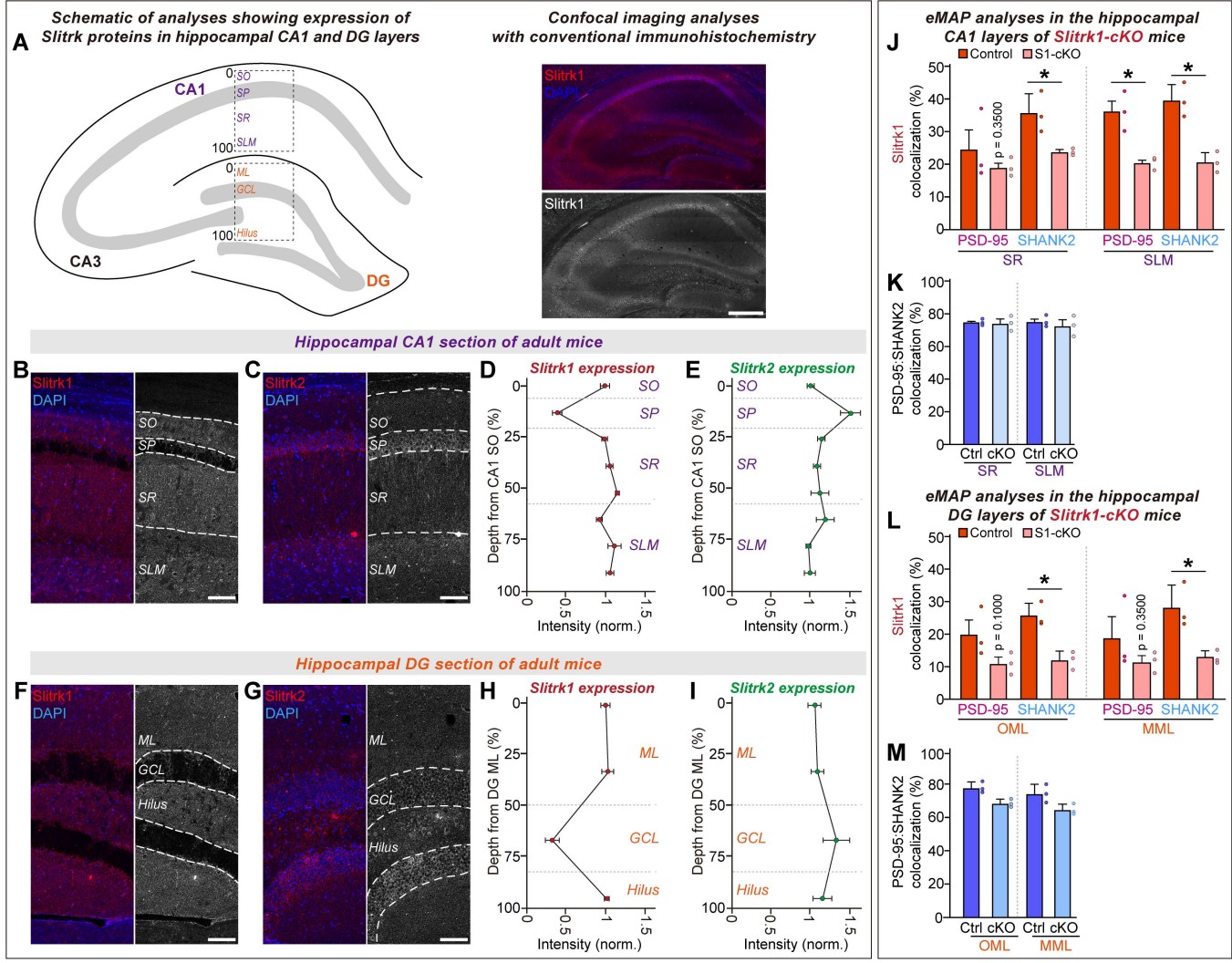

**Fig 1. Slitrk1 and Slitrk2 proteins are widely expressed in various layers of hippocampal CA1 and DG but exhibit distinct enriched expression patterns. (A)** Schematic of immunohistochemical analysis strategy used to analyze the protein expression patterns of Slitrk1 and Slitrk2 in the hippo-campal subfields of adult mice. Representative images showing Slitrk1 protein distribution in coronal brain sections from adult male mice. Scale bar, 100 μm (applies to all images). **(B–I)** Representative images (B, F for Slitrk1 and C, G for Slitrk2) and quantification of mean intensity distributions for Slitrk1 (D, H) and Slitrk2 (E, I) across CA1 and DG layers. Data are presented as means±SEMs (*n*=6 mice/group). Scale bars, 50 μm (applies to all images). Abbreviations: SO, *stratum oriens*; SR, *stratum radiatum*; SP, *stratum pyramidale*; SLM, *stratum lacunosum-moleculare*; GCL, granule cell layer; ML, molecular layer. **(J and L)** Quantitative analysis of the colocalization of Slitrk1 with the postsynaptic markers, SHANK2 and PSD-95, in the SR and SLM layers of the hippocampal CA1 (J) and the OML and MML layers of the DG (L). Data are presented as means±SEMs (*n*=3 mice/group; **p*<0.05; one-tailed Mann–Whitney *U* test). **(K and M)** Quantitative analysis of the colocalization of the excitatory synapse markers, SHANK2 and PSD-95 in the SR and SLM layers of the hippocampal CA1 (K) and the OML and MML layers of the DG (M). This is a staining quality control in which cKO should not alter colocalization. Colocalization was measured from three separate gels for each mouse, and values were averaged for each mouse. Data are presented as means±SEMs (*n*=3 mice/group; **p*<0.05; one-tailed Mann–Whitney *U* test). Numerical data can be found in S1 Data.

layers of the CA1 and DG: Slitrk1 is more prominently expressed beyond the *stratum pyramidale* (SP) and granular cell layers, whereas Slitrk2 is broadly expressed throughout the cell layers (Fig 1A–1I).

To further analyze the expression patterns of Slitrk1 at nanoscopic resolution, we performed eMAP analysis [32] (Fig 1J–1Q; Slitrk antibodies were again validated using the corresponding *Slitrk1*-cKO mice; S2 Fig). eMAP protocols enable maximal

preservation of native epitopes in conjunction with tissue expansion-based super-resolution microscopic analyses. When seeking to validate the Slitrk1 and Slitrk2 antibodies in the eMAP-processed tissues, we found that the total number of Slitrk1+ puncta was reduced in the *Slitrk1*-cKO samples compared to controls (S2 Fig). However, the Slitrk2+ puncta number was comparable between the control and *Slitrk2*-cKO groups (S2 Fig), indicating that the Slitrk2 antibody was not applicable in the eMAP-processed tissues. For Slitrk1, we analyzed colocalization with postsynaptic density 95 (PSD-95) and SHANK2, both of which are markers for excitatory postsynaptic sites. Our results revealed that Slitrk1 exhibited regional differences in their colocalization with SHANK2 and PSD-95 (Figs 1J–1Q and S2). Slitrk1 protein colocalized more highly with SHANK2 and PSD-95 in the hippocampal SR and SLM layers of the CA1 compared to the outer molecular layer (OML) and medial molecular layer (MML) of the DG (Figs 1J, 1N, and S2). The colocalization patterns for Slitrk1 were comparable between the OML and MML, with no marked differences in their colocalizations with SHANK2 or PSD-95 (Figs 1L and S2). Collectively, these results indicate that Slitrk1 exhibits distinct expression patterns across different hippocampal layers and show enriched colocalization with excitatory synaptic markers.

## Slitrk1 and Slitrk2 exert opposite regulatory influences on basal excitatory synaptic transmission in hippocampal CA1 pyramidal neurons

A previous study demonstrated that Slitrk1 differentially controls synaptic organization in hippocampal CA1 synapses in an input-specific manner [18]. However, this study utilized a short hairpin-based KD approach [18], leaving open the possibility of off-target effects. Moreover, parallel Slitrk2 loss-of-function analyses were not performed. Thus, we generated *Slitrk1* conditional knockout (*Slitrk1*-cKO) mice in which exon 1 was deleted through Cre recombinase-dependent excision at flanking loxP sites (*Slitrk1*f/f) (S1A and S1B Fig). After confirming that *Slitrk1*-cKO mice were properly generated using quantitative RT-PCR analyses (S1C Fig), we examined expression levels of synaptic proteins in these mice as well as previously established *Slitrk2*-cKO mice [17]. We found that expression levels of most examined synaptic proteins were not altered in either *Slitrk1*-cKO or *Slitrk2*-cKO mice (S1E and S1F Fig).

We then injected AAVs expressing Cre recombinase (Cre) or inactive Cre (ΔCre) into the hippocampal CA1 of adult *Slitrk1*f/f and *Slitrk2*f/f mice and performed recordings of CA1 pyramidal neurons from control and the respective *Slitrk*-cKO mice 2 weeks after the injections (Fig 2A and 2B). We confirmed the correct targeting of AAVs and prominent protein downregulation of each Slitrk in the AAV-infected neurons of the hippocampal CA1 (S3 Fig; note that the AAV infection efficiency was around ~95%). We then measured the action potential (AP) firing rate of CA1 pyramidal neurons induced by current injections in acute CA1 slices from *Slitrk1*-cKO, *Slitrk2*-cKO and control mice. We found that the AP firing rate was increased in *Slitrk1*-cKO CA1 neurons, but was decreased in *Slitrk2*-cKO CA1 neurons (S4A–S4D Fig). We next measured spontaneous excitatory postsynaptic currents (sEPSCs). *Slitrk1*-cKO CA1 pyramidal neurons displayed increased frequency, but not amplitude, of sEPSCs (Fig 2C and 2D), in line with the phenotype of *Slitrk1*-KD rats [18] (see S2 Table). Conversely, parallel experiments using *Slitrk2*-cKO CA1 pyramidal neurons showed a decrease in sEPSC frequency (Fig 2E and 2F), in keeping with our previous report [17]. However, the similar trends of change in excitability and sEPSCs in CA1 pyramidal neurons of each *Slitrk*-cKO led us to ask whether the changes in sEPSCs could reflect alterations of intrinsic properties. Thus, we employed designer receptors that were exclusively activated by designer drug (DREADD)-based chemogenetics (S5A Fig). We injected adult *Slitrk1*f/f or *Slitrk2*f/f mice with AAVs expressing Cre or ΔCre, together with AAVs expressing inhibitory (AAV-hM4Di) or excitatory (AAV-hM3Dq) DREADDs, and further applied deschloroclozapine (DCZ) or saline 1 hour prior to electrophysiological recordings (S5A Fig). We validated the DCZ-induced normalization of DREADD receptors by measuring firing rates in the CA1 pyramidal neurons from *Slitrk1*-cKO or *Slitrk2*-cKO mice (S5 Fig). The CA1 pyramidal neurons expressing hM4Di and hM3Dq exhibited decreased and increased firing rates, respectively, after DCZ was applied to brain sections from *Slitrk1*-cKO or *Slitrk2*-cKO mice (S5 Fig). We then examined whether the normalized firing rates of *Slitrk*-cKO CA1 pyramidal neurons were associated with alterations of sEPSCs. There was no change in the sEPSC phenotypes of *Slitrk*-cKO mouse-derived CA1 pyramidal neurons whose excitabilities

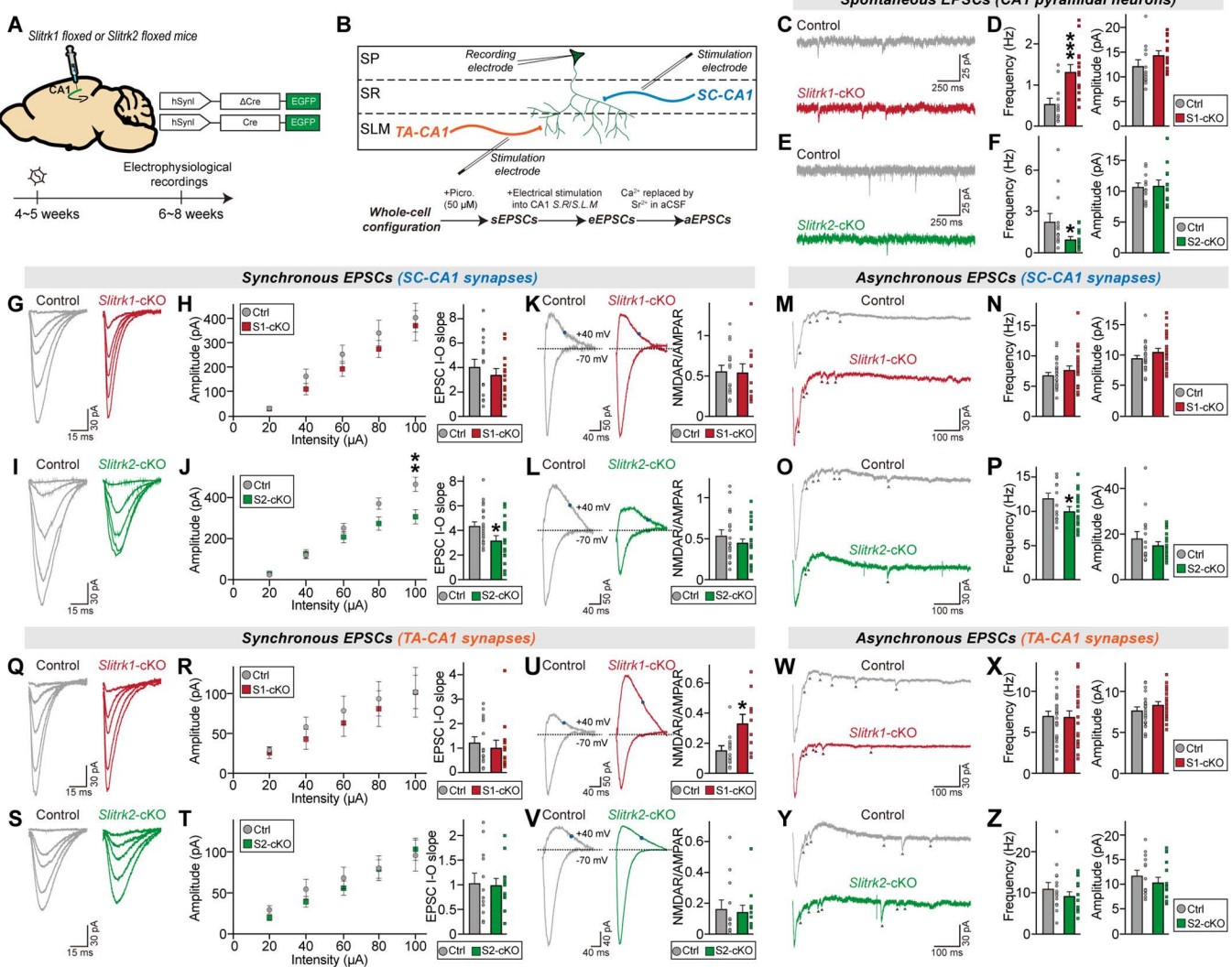

**Fig 2. Slitrk1 and Slitrk2 differentially control synaptic transmission in the hippocampal CA1 in a microcircuit-dependent fashion. (A)** Schematic showing AAV injection into the hippocampal CA1 of *Slitrk1*^f/f and *Slitrk2*^f/f mice. Electrophysiological recordings were performed 2 weeks after injections. **(B)** Schematic showing whole-cell recordings from CA1 pyramidal neurons. **(C–F)** Whole-cell recordings of sEPSCs from CA1 pyramidal neurons in control (gray), *Slitrk1*-cKO (red), and *Slitrk2*-cKO (green) hippocampal slices. Representative traces (C and E) and averages of sEPSCs (D and F; control in (D), $n = 12/4$; control in (F), $n = 13/4$; *Slitrk1*-cKO, $n = 14/4$; *Slitrk2*-cKO, $n = 13/4$, where '$n$' denotes the number of cells/mice). **(G–J)** Whole-cell recordings of eEPSCs at SC-CA1 synapses in control, *Slitrk1*-cKO, and *Slitrk2*-cKO hippocampal slices. Representative traces (G and I) and averages of eEPSCs (H and J; control in (H), $n = 17/5$; control in (J), $n = 22/7$; *Slitrk1*-cKO, $n = 13/5$; *Slitrk2*-cKO, $n = 21/7$). **(K and L)** Representative traces of NMDAR/AMPAR-EPSC at SC-CA1 synapses in control, *Slitrk1*-cKO, and *Slitrk2*-cKO hippocampal slices (left) and averages of NMDAR/AMPAR-EPSCs (K and L; control in (K), $n = 17/5$; control in (L), $n = 19/7$; *Slitrk1*-cKO, $n = 12/5$; *Slitrk2*-cKO, $n = 21/7$). **(M–P)** Whole-cell recordings of aEPSCs at SC-CA1 synapses in control, *Slitrk1*-cKO, and *Slitrk2*-cKO hippocampal slices. Representative traces (M and O) and averages of eEPSCs (N and P; control in (N), $n = 28/7$; control in (P), $n = 16/5$; *Slitrk1*-cKO, $n = 27/7$; *Slitrk2*-cKO, $n = 18/5$). **(Q–T)** Whole-cell recordings of eEPSCs at TA-CA1 synapses in control, *Slitrk1*-cKO, and *Slitrk2*-cKO hippocampal slices. Representative traces (Q and S) and averages of eEPSCs (R and T; control in (R), $n = 14/5$; control in (T), $n = 12/5$; *Slitrk1*-cKO, $n = 13/5$; *Slitrk2*-cKO, $n = 13/5$). **(U and V)** Representative traces of NMDAR/AMPAR-EPSC at TA-CA1 synapses in control, *Slitrk1*-cKO, and *Slitrk2*-cKO hippocampal slices (left) and averages of NMDAR/AMPAR-EPSCs (control in (U), $n = 14/5$; control in (V), $n = 12/5$; *Slitrk1*-cKO, $n = 11/5$; *Slitrk2*-cKO, $n = 12/5$). **(W–Z)** Whole-cell recordings of aEPSCs at TA-CA1 synapses in control, *Slitrk1*-cKO, and *Slitrk2*-cKO hippocampal slices. Representative traces (W and Y) and averages of eEPSCs (X and Z; control in (X), $n = 27/7$; control in (Z), $n = 14/5$; *Slitrk1*-cKO, $n = 25/7$; *Slitrk2*-cKO, $n = 16/5$). Data are presented as means ± SEMs (*$p < 0.05$, **$p < 0.01$, ***$p < 0.001$; two-tailed nonparametric Mann–Whitney $U$ test). Numerical data can be found in S1 Data.

had been normalized by DREADDs (S6 Fig), suggesting that the intrinsic property alterations induced by conditional Slitrk ablation do not influence basal synaptic transmission. Collectively, our results unambiguously established opposing roles of two Slitrk paralogs in regulating basal synaptic transmission at excitatory synapses.

### Slitrk1 and Slitrk2 distinctively regulate evoked synaptic strength at different excitatory synaptic inputs in hippocampal CA1 pyramidal neurons

Because there are three primary modes of neurotransmitter release at synapses—synchronous, asynchronous, and spontaneous [38]—we next measured evoked excitatory postsynaptic currents (eEPSCs) in CA1 pyramidal neurons from control and respective *Slitrk*-cKO mice (Fig 2B). To discriminate the contribution of Slitrks in mediating evoked synaptic strength at different excitatory inputs, we electrically stimulated axonal fibers of Schaffer collateral (SC)-CA1 or tempo-roammonic (TA)-CA1 synapses, which target the SR and SLM layers of the CA1, respectively, and then recorded the eEPSCs of CA1 pyramidal neurons using input-out (I-O) curves to control for the variability in stimulus strength.

Postsynaptic Slitrk1 deletion at SC-CA1 synapses did not affect any examined synaptic properties (Figs 2G, 2K, 2H, S7A and S7B). In contrast, loss of postsynaptic Slitrk2 significantly decreased the amplitude of eEPSCs at SC-CA1 synapses (Fig 2I, 2J, S7C and S7D), which exclusively reflect α-amino-3-hydroxyl-5-methylisoxazole-4-prionic acid receptor (AMPAR)-mediated responses. However, parallel experiments assessing N-methyl-D-aspartate receptor (NMDAR)-mediated responses also showed a decrease in synaptic strength, such that the NMDAR/AMPAR ratio was comparable between control and *Slitrk2*-cKO neurons (Fig 2L), indicating that NMDAR- and AMPAR-mediated responses are reduced in Slitrk2-deficient CA1 pyramidal neurons. Postsynaptic Slitrk1 deletion at TA-CA1 synapses also induced no alterations in AMPAR-EPSCs or AMPAR-EPSC-paired pulse ratios (PPRs), a proxy for neurotransmitter release probability (Figs 2Q, 2R, S7E and S7F). Strikingly, the loss of synaptic Slitrk1 in CA1 pyramidal neurons increased the amplitude of NMDAR-EPSCs at TA-CA1 synapses, as also reflected in the increased the NMDAR/AMPAR ratio (Fig 2U). In contrast, postsynaptic Slitrk2 deletion at TA-CA1 synapses did not alter any examined synaptic properties (Figs 2S–2V, S7G and S7H).

We next measured asynchronous EPSCs (aEPSCs), given their emerging role in shaping synaptic efficacy, neurotransmission reliability and plasticity [39], in *Slitrk*-cKO CA1 pyramidal neurons. To induce delayed synaptic transmission, we replaced $Ca^{2+}$ with $Sr^{2+}$ in the extracellular solution for the recordings and electrically stimulated axon fibers originating from the CA3 or entorhinal cortex, respectively, to measure aEPSCs at the corresponding SC-CA1 and TA-CA1 circuits (Fig 2M–2P and 2W–2Z). Postsynaptic Slitrk1 deletion did not induce any discernible changes in the frequency or amplitude of aEPSCs at either SC-CA1 or TA-CA1 synapses (Fig 2M, 2N, 2W, and 2X). In contrast, loss of postsynaptic Slitrk2 significantly decreased the frequency (but not amplitude) of aEPSCs at SC-CA1 (but not TA-CA1) synapses (Fig 2O, 2P, 2Y, and 2Z). Collectively, our data suggest that different excitatory Slitrk paralogs differentially regulate the synaptic composition of glutamate receptors at distinct excitatory inputs onto CA1 pyramidal neurons.

### Slitrk1 and Slitrk2 distinctively regulate evoked synaptic strength at different excitatory synaptic inputs in hippocampal dentate gyrus granule neurons

We next explored the question of whether two Slitrk paralogs can distinctively regulate synaptic properties at different excitatory synaptic inputs in a circuit context-dependent manner. To address this question, we shifted our attention to DG granule neurons, which highly express *Slitrk* mRNAs [40]. Adult *Slitrk1*^f/f and *Slitrk2*^f/f mice were injected with AAV-Cre or AAV-ΔCre (control) and sEPSCs were recorded (Fig 3A and 3B). Unexpectedly, we found that sEPSCs were unaltered in DG granule neurons from both *Slitrk1*-cKO and *Slitrk2*-cKO mice (Fig 3C–3F). Moreover, the AP firing rate was unchanged in *Slitrk1*-cKO and *Slitrk2*-cKO DG granule neurons compared with control neurons (S4E–S4H Fig). These data suggest that neither Slitrk1 nor Slitrk2 is required for maintenance of basal excitatory synaptic transmission in adult DG granule neurons, in contrast to their roles in CA1 pyramidal neurons.

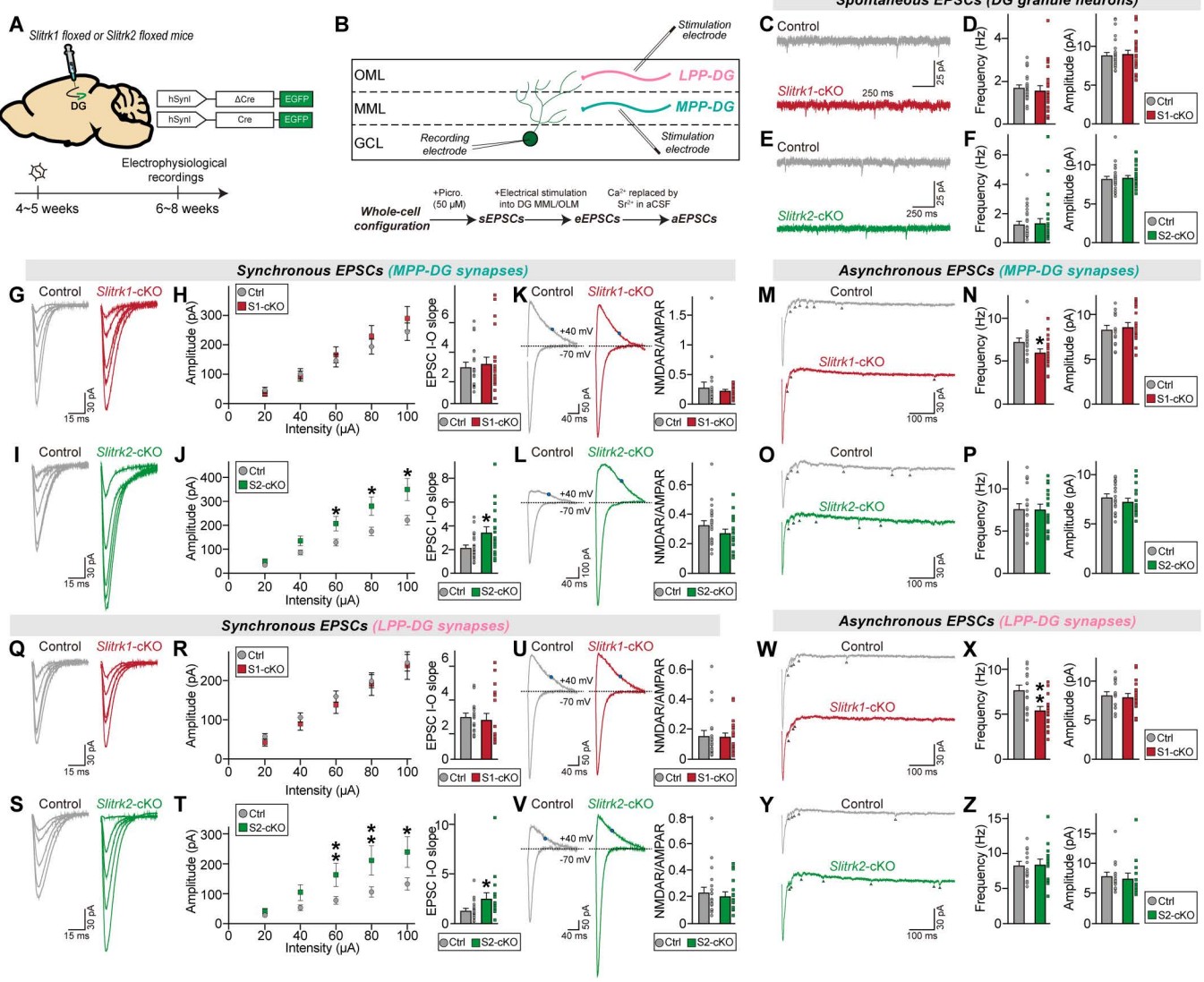

**Fig 3. Slitrk1 and Slitrk2 differentially control synaptic transmission in the hippocampal DG in a microcircuit-dependent manner. (A)** Schematic showing AAV injection into the hippocampal DGs of *Slitrk1*^f/f and *Slitrk2*^f/f mice. Electrophysiological recordings were performed 2 weeks after injections. **(B)** Schematic showing whole-cell recordings from DG granule neurons. **(C–F)** Whole-cell recordings of sEPSCs from DG granule neurons of control (gray), *Slitrk1*-cKO (red), and *Slitrk2*-cKO (green) hippocampal slices. Representative traces (C and E) and averages of sEPSCs (D and F; control in (D), n=29/7; Control in (F), n=27/7; *Slitrk1*-cKO, n=25/7; *Slitrk2*-cKO, n=27/7, where 'n' denotes number of cells/mice). **(G–J)** Whole-cell recordings of eEPSCs at MPP-DG synapses of control, *Slitrk1*-cKO, and *Slitrk2*-cKO hippocampal slices. Representative traces (G and I) and averages of eEPSCs (H and J; control in (H), n=20/7; control in (J), n=23/7; *Slitrk1*-cKO, n=21/7; *Slitrk2*-cKO, n=20/7). **(K and L)** Representative traces of NMDAR/AMPAR-EPSCs at MPP-DG synapses of control, *Slitrk1*-cKO, and *Slitrk2*-cKO hippocampal slices (left) and averages of NMDAR/AMPAR-EPSCs (control in (K), n=20/7; control in (L), n=23/7; *Slitrk1*-cKO, n=21/7; *Slitrk2*-cKO, n=20/7). **(M–P)** Whole-cell recordings of aEPSCs at MPP-DG synapses of control, *Slitrk1*-cKO, and *Slitrk2*-cKO hippocampal slices. Representative traces (M and O) and averages of eEPSCs (N and P; control in (N), n=16/7; control in (P), n=20/7; *Slitrk1*-cKO, n=19/7; *Slitrk2*-cKO, n=18/7). **(Q–T)** Whole-cell recordings of eEPSCs at LPP-DG synapses of control, *Slitrk1*-cKO, and *Slitrk2*-cKO hippocampal slices. Representative traces (Q and S) and averages of eEPSCs (R and T; control in (R), n=20/7; control in (T), n=21/7; *Slitrk1*-cKO, n=20/7; *Slitrk2*-cKO, n=17/7). **(U and V)** Representative traces of NMDAR/AMPAR-EPSCs at LPP-DG synapses of control, *Slitrk1*-cKO, and *Slitrk2*-cKO hippocampal slice (left) and averages of NMDAR/AMPAR-EPSCs (control in (U), n=20/7; control in (V), n=20/7; *Slitrk1*-cKO, n=21/7; *Slitrk2*-cKO, n=17/7). **(W–Z)** Whole-cell recordings of aEPSCs at LPP-DG synapses of control, *Slitrk1*-cKO, and *Slitrk2*-cKO hippocampal slices. Representative traces (W and Y) and averages of aEPSCs (X and Z; control in (X), n=16/7; control in (Z), n=15/7; *Slitrk1*-cKO, n=19/7; *Slitrk2*-cKO, n=14/7). Data are presented as means±SEMs (*p<0.05, **p<0.01; two-tailed nonparametric Mann–Whitney *U* test). Numerical data can be found in S1 Data.

We then asked whether Slitrk1 and/or Slitrk2 is involved in the regulation of evoked excitatory synaptic strength at specific synapses in DG granule neurons. To measure eEPSCs and aEPSCs, we electrically stimulated axons of the medial perforant path (MPP) or lateral PP (LPP), which target the MML and OML of the DG, respectively, and then recorded eEPSCs and aEPSCs of DG granule neurons. The results are presented in the form of I-O curves as done for CA1 pyramidal neurons (see Fig 2). Postsynaptic Slitrk1 deletion in DG granule neurons did not affect AMPAR-EPSCs, NMDAR-EPSCs or PPRs at either MPP-DG or LPP-DG synapses (Figs 3G, 3H, 3K, 3Q, 3R, 3U, S7I, S7J, S7M, and S7N). Conversely, loss of postsynaptic Slitrk2 increased the amplitudes of both AMPAR-EPSCs and NMDAR-EPSCs without altering PPRs in adult DG granule neurons, irrespective of synapse identities (Figs 3I, 3J, 3L, 3S, 3T, 3V, S7K, S7L, S7O, and S7P). These observations suggest that Slitrk2 negatively regulates the synaptic composition of glutamate receptors in DG granule neurons, a function distinct from its role as a synaptic facilitator at SC-CA1 synapses.

Measurement of aEPSCs at MPP-DG and LPP-DG synapses in Slitrk-deficient DG granule neurons by electrically stimulating axons at 20 Hz revealed that postsynaptic Slitrk1 deletion in DG granule neurons decreased aEPSC frequency at both MPP-DG and LPP-DG synapses (Fig 3M, 3N, 3W, and 3X). In contrast, postsynaptic Slitrk2 deletion in DG granule neurons did not produce any changes in aEPSCs at the examined DG synapses (Fig 3O, 3P, 3Y, and 3Z). Taken together, these findings indicate that both excitatory Slitrk paralogs function differentially in regulating distinct excitatory synaptic properties in DG granule neurons, similar to the case in CA1 pyramidal neurons, but do so in a microcircuit-independent manner.

## Deletion of Slitrk1 or Slitrk2 decreases excitatory synapse numbers in the hippocampal CA1, but not DG, of adult mice

Given the distinct phenotypic outcomes observed in hippocampal CA1 and DG neurons from each *Slitrk*-cKO mouse line, we investigated whether the loss of Slitrk1 or Slitrk2 in the hippocampal CA1 or DG could alter excitatory synapse density. We stereotactically injected AAVs expressing Cre or ΔCre into the CA1 or DG of adult *Slitrk*-floxed mice. Two weeks post-injection, we performed semi-quantitative immunohistochemistry (S8 Fig). Consistent with the results of our previous study [17], the injected *Slitrk2*f/f mice exhibited significant reductions in the integrated intensities of VGLUT1+PSD-95+ puncta in the SO and SR layers, but not in the SLM layer (S8 Fig). Intriguingly, injected *Slitrk1*f/f mice also displayed a similar significant reduction in the integrated intensity of VGLUT1+PSD-95+ puncta in the hippocampal CA1 layers (S8 Fig). This result appeared to contradict the increased sEPSC frequency observed in CA1 pyramidal neurons (see Fig 2). To resolve this potential discrepancy, we performed electrophysiological recordings to measure mEPSCs in CA1 pyramidal neurons from adult *Slitrk1*-cKO mice. We found that the frequency, but not amplitude, of mEPSCs was markedly decreased in hippocampal CA1-specific *Slitrk1*-cKO mice (S9 Fig). This phenotype was similar to that previously reported for hippocampal CA1-specific *Slitrk2*-cKO mice [17] and reproduced in the current study (S9 Fig).

In contrast, hippocampal DG-specific *Slitrk1*-cKO, but not *Slitrk2*-cKO, mice exhibited a significant reduction in the integrated intensity of VGLUT1+PSD-95+ puncta in the molecular layer (S8 Fig), suggesting that Slitrk1 contributes to maintaining the excitatory synapse density in the hippocampal DG, but not CA1, of adult mice. This finding appears incongruous with the lack of change in sEPSCs, but might reflect changes in other excitatory synaptic parameters in DG granule neurons of *Slitrk1*-cKO mice (see Fig 3). Taken together, our results suggest that both Slitrk1 and Slitrk2 are crucial for maintaining excitatory synapse density in the CA1, and Slitrk1 is further involved in maintaining excitatory synapse density in the DG area of adult mice.

## Deletion of Slitrk1 or Slitrk2 produces marginal but distinct alterations in presynaptic vesicle organization

We next investigated whether the synaptic ultrastructure within the diverse hippocampal circuits is altered in *Slitrk1*-cKO or *Slitrk2*-cKO mice (Figs 4 and S10). To this end, we injected AAVs expressing ΔCre or Cre into the hippocampal CA1 or DG of adult *Slitrk1*f/f and *Slitrk2*f/f mice and obtained images of presynaptic boutons at different layers in the CA1 or DG by capturing transmission electron microscopy (TEM) images along the axis of DG–to–CA1 dendrites (Figs 4A and S10A).

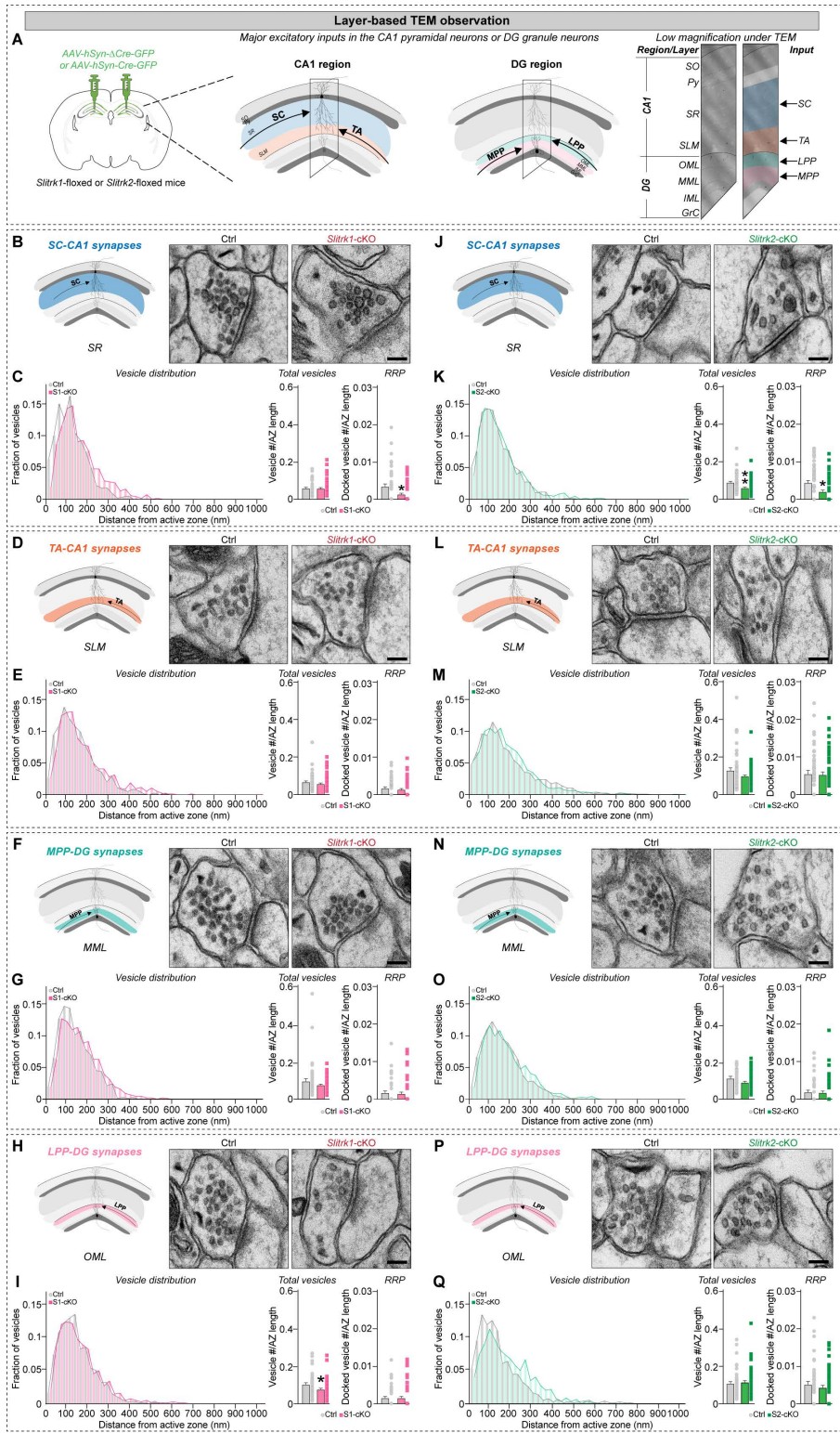

**Fig 4. Slitrk1 and Slitrk2 exert differential effects on vesicle density within diverse hippocampal circuits. (A)** Schematic demonstrating input-specific observations of the CA1 and DG regions using transmission electron microscopy. **(B, D, F, and H)** Diagram illustrating the observed layers and representative electron micrographs of synapses in CA1 SR (B), CA1 SLM (D), DG MML (F) and DG OML (H) layers from control and *Slitrk1*-cKO mice.

Scale bars, 100 nm (applies to all images). **(C, E, G, and I)** Quantification of vesicle distribution by distance from the active zone (AZ) (left; expressed in 25 nm bins as the relative frequency distribution of vesicles), total vesicle density (middle; number of vesicles normalized to AZ length), and docked vesicle density (right; number of docked vesicles normalized to AZ length) at SC-CA1 (C), TA-CA1 (E), MPP-DG (G) and LPP-DG (I) synapses of *Slitrk1*-cKO compared with control mice. **(J, L, N, and P)** Diagram illustrating the observed layers and representative electron micrographs of synapses in CA1 SR (J), CA1 SLM (L), DG MML (N) and DG OML (P) layers from control and *Slitrk2*-cKO mice. Scale bars, 100 nm (applies to all images). **(K, M, O, and Q)** Quantification of vesicle distribution by distance from the AZ (left), total vesicle density (middle), and docked vesicle density (right) at SC-CA1 (K), TA-CA1 (M), MPP-DG (O), and LPP-DG (Q) synapses of *Slitrk2*-cKO compared with control mice. Data represents means ± SEMs (control, $n = 45/3$; *Slitrk1*-cKO or *Slitrk2*-cKO, $n = 60/4$, where $n$ denotes the number of synapses/mice; *$p < 0.05$, **$p < 0.01$; n.s., not significant; Kolmogorov–Smirnov test). Numerical data can be found in S1 Data.

Specifically, we examined the MML and OML encompassing MPP and LPP, respectively, and SR and SLM layers encompassing SC and TA inputs, respectively (Fig 4A). We found no changes in AZ length in any examined layer of hippocampal subfields in *Slitrk1*-cKO mice compared with the respective control mice (S10C–S10F Fig) but did detect a paradoxical increase in the length of the postsynaptic density (PSD) exclusively at the SR layer of *Slitrk1*-cKO mice (S10C Fig). In addition, the widths of synaptic clefts in the SR and SLM layers of the CA1 were slightly, but not significantly, greater in *Slitrk1*-cKO than in control mice (S10C and S10D Fig). A distance map of the AZ created to analyze vesicle location and its distribution (S10B Fig) revealed a significant decrease in the density of total vesicles at the OML of the DG in *Slitrk1*-cKO mice (Fig 4H and 4I; summarized in S3 Table). However, the density of docked vesicles, defined as those less than 25 nm from the AZ, was significantly lower in the SR layer of the CA1 in *Slitrk1*-cKO mice than control mice (Fig 4B and 4C). The density of docked vesicles in DG layers was not different between control and *Slitrk1*-cKO mice (Fig 4F–4I). These results suggest a role for Slitrk1 in the proper maintenance of the readily releasable pool in the CA1 SR layer.

To assess whether Slitrk2 acts similarly in regulating the ultrastructure of hippocampal synapses, we performed similar TEM analyses in these mice. Unexpectedly, the synaptic structures, such as the AZ, PSD, and synaptic cleft, were unchanged in all examined layers of the hippocampus in *Slitrk2*-cKO mice (S10G–S10J Fig). We found that the number of total and docked vesicles in the SR layer of the CA1 was significantly reduced in *Slitrk2*-cKO mice compared to control mice, with no alterations in vesicle distribution within the bouton (Fig 4J and 4K). These data indicate that Slitrk2 is specifically required for maintenance of the number of vesicles in SC-CA1 synapses. Additionally, both total and docked vesicle densities were comparable in the DGs of control and *Slitrk2*-cKO mice (Fig 4N–4Q). Collectively, these ultrastructural analyses reveal that Slitrk1 and Slitrk2 contribute differentially to synaptic vesicle organization at distinct inputs in the hippocampal CA1 and DG.

### The actions of Slitrk1 and Slitrk2 in shaping excitatory synaptic properties in hippocampal CA1 pyramidal neurons and DG granule neurons are not redundant

Extensive electrophysiological analyses suggest that both Slitrk1 and Slitrk2 perform distinct functions in regulating various synaptic properties in the hippocampal CA1 and DG. However, there are precedents for functional redundancy among members of the same CAM family [28, 41–44]. Moreover, a subset of Slitrk synaptic functions might be occluded by functional redundancy, masking a subset of physiological effects on circuit properties. Thus, we established *Slitrk1*ᶠ/ᶠ/*Slitrk2*ᶠ/ᶠ (*Slitrk1/2*-dKO) mice, validated by quantitative immunoblotting (see S1E and S1F Fig), and performed similar electrophysiological recordings in hippocampal CA1 pyramidal neurons or DG granule neurons to compare the phenotypes of *Slitrk1/2*-dKO mice with those of single *Slitrk*-cKO mice (S11A Fig). We found that *Slitrk1/2*-dKO neurons exhibited impairments in synaptic transmission similar to those in *Slitrk1*-cKO and *Slitrk2*-cKO neurons (S11 Fig), namely decreased amplitude of eEPSCs and decreased frequency of aEPSCs at SC-CA1 synapses, increased NMDAR/AMPAR ratio at TA-CA1 synapses, increased amplitude of eEPSCs at both MPP-DG and LPP-DG synapses, and decreased frequency of aEPSCs at both MPP-DG and LPP-DG synapses (S11F–S11Y Fig and S4 Table). However, *Slitrk1/2*-dKO neurons exhibited excitability, PPR and sEPSC phenotypes comparable to those of control neurons, which contrasted with the distinct

phenotypes of each *Slitrk*-cKO neurons (S4I–S4L, S7Q–S7T, and S11B–S11E Fig). These results reveal that opposite phenotypes are produced by loss of each Slitrk paralog, suggesting that Slitrk1 and Slitrk2 are functionally nonredundant in regulating various excitatory synaptic properties in the hippocampal CA1 and DG microcircuits; instead, they act via distinct and opposing mechanisms to regulate basal synaptic transmission and intrinsic excitability in CA1 pyramidal neurons, as clearly demonstrated by the opposite phenotypes produced by the loss of each Slitrk paralog.

**Slitrk1 and Slitrk2 deploy distinct extracellular mechanisms to control excitatory synaptic properties**

How do Slitrk1 and Slitrk2 differentially shape excitatory synaptic properties across hippocampal circuits? The extracellular domains of Slitrk proteins are structurally similar across family members and bind in common to leukocyte common antigen related receptor protein tyrosine phosphatases (LAR-RPTPs) to induce presynaptic assembly [12,45–47]. However, a recent study demonstrated a circuit-dependent role for PTPδ, a member of the LAR-RPTP family, in regulating excitatory synaptic transmission in the hippocampal circuits [48], prompting us to first examine whether the LAR-RPTP–binding activity of Slitrk1 and Slitrk2 is involved in executing their synaptic functions. To this end, we designed Slitrk variants Slitrk1 N162R (Slitrk1 PBM) and Slitrk2 N166R (Slitrk2 PBM), based on the published crystal structures of Slitrk1-PTPδ [49], to specifically disrupt their binding to LAR-RPTPs (Fig 5A). Both Slitrk point mutants were expressed well on the surface of transfected human embryonic kidney 293T (HEK293T) cells and showed the expected disruption of binding to PTPσ-Fc proteins (S12 Fig). Moreover, Slitrk PBM variants exhibited expression levels and excitatory synaptic targeting efficacies comparable to those of their WT counterparts when overexpressed in cultured hippocampal neurons (S13 Fig).

*Slitrk1*^f/f^ and *Slitrk2*^f/f^ mice were then injected with AAV-Cre, or co-injected with AAV-Cre and AAVs expressing the respective Slitrk wild-type (WT) or Slitrk PBM, and then electrophysiological recordings were performed to determine if the expression of Slitrk PBM could rescue the abnormalities observed in each type of *Slitrk*-cKO neurons. Four synaptic parameters—sEPSCs in CA1 pyramidal neurons, NMDAR-EPSCs at TA-CA1 synapses, aEPSCs at MPP-DG synapses, and aEPSCs at LPP-DG synapses—were monitored in hippocampal circuits of *Slitrk1*-cKO mice (Fig 5B). Expression of Slitrk1 WT completely rescued the abnormalities in all examined synaptic properties of *Slitrk1*-cKO neurons, whereas expression of Slitrk1 PBM failed to do so (Fig 5C–5J), indicating that Slitrk1 requires extracellular interactions with LAR-RPTPs to dictate excitatory synaptic properties in both CA1 and DG neurons. We then monitored four synaptic parameters—sEPSCs in CA1 pyramidal neurons, eEPSCs and aEPSCs at SC-CA1 synapses, eEPSCs at MPP-DG synapses, and eEPSCs at LPP-DG synapses—in the hippocampal circuits of *Slitrk2*-cKO mice (Fig 5B). Notably, excluding eEPSCs at SC-CA1 synapses, all abnormalities in examined synaptic properties of *Slitrk2*-cKO neurons were reversed by expression of Slitrk2 PBM, similar to expression of Slitrk2 WT (Fig 5K–5Z). Collectively, these results suggest that both Slitrk paralogs require LAR-RPTP-binding activity to reciprocally regulate basal synaptic transmission, and that LAR-RPTPs engage with Slitrk1, but not Slitrk2, in regulating the specification of excitatory synaptic properties (except for eEPSCs at SC-CA1 synapses) in hippocampal microcircuits.

**PDZ domain-containing proteins are required for Slitrk2 regulation of spontaneous, but not synchronous or asynchronous, excitatory synaptic transmission in hippocampal circuits**

Although the interaction of Slitrk2 with LAR-RPTPs is required to control eEPSCs at SC-CA1 synapses, the other excitatory synaptic properties governed by Slitrk2 are not dictated by this interaction. In contrast to Slitrk1, Slitrk2 interacts with TrkB and PDZ domain-containing proteins [14,17,50]. Thus, we next asked whether the ability of Slitrk2 to shape synaptic properties, which does not appear to be dictated by its LAR-RPTP-binding activity, depends on its binding to TrkB and PDZ proteins. To this end, we generated AAVs expressing Slitrk2 ΔPDZ, a Slitrk2 variant lacking the C-terminal four amino acids, Ile-Ser-Glu-Leu [14] [note that Slitrk2 ΔPDZ also exhibited expression levels and excitatory synaptic targeting efficacies comparable to those of Slitrk2 WT (see S13 Fig)]. We then co-injected *Slitrk2*^f/f^ mice with AAV-Cre and AAVs

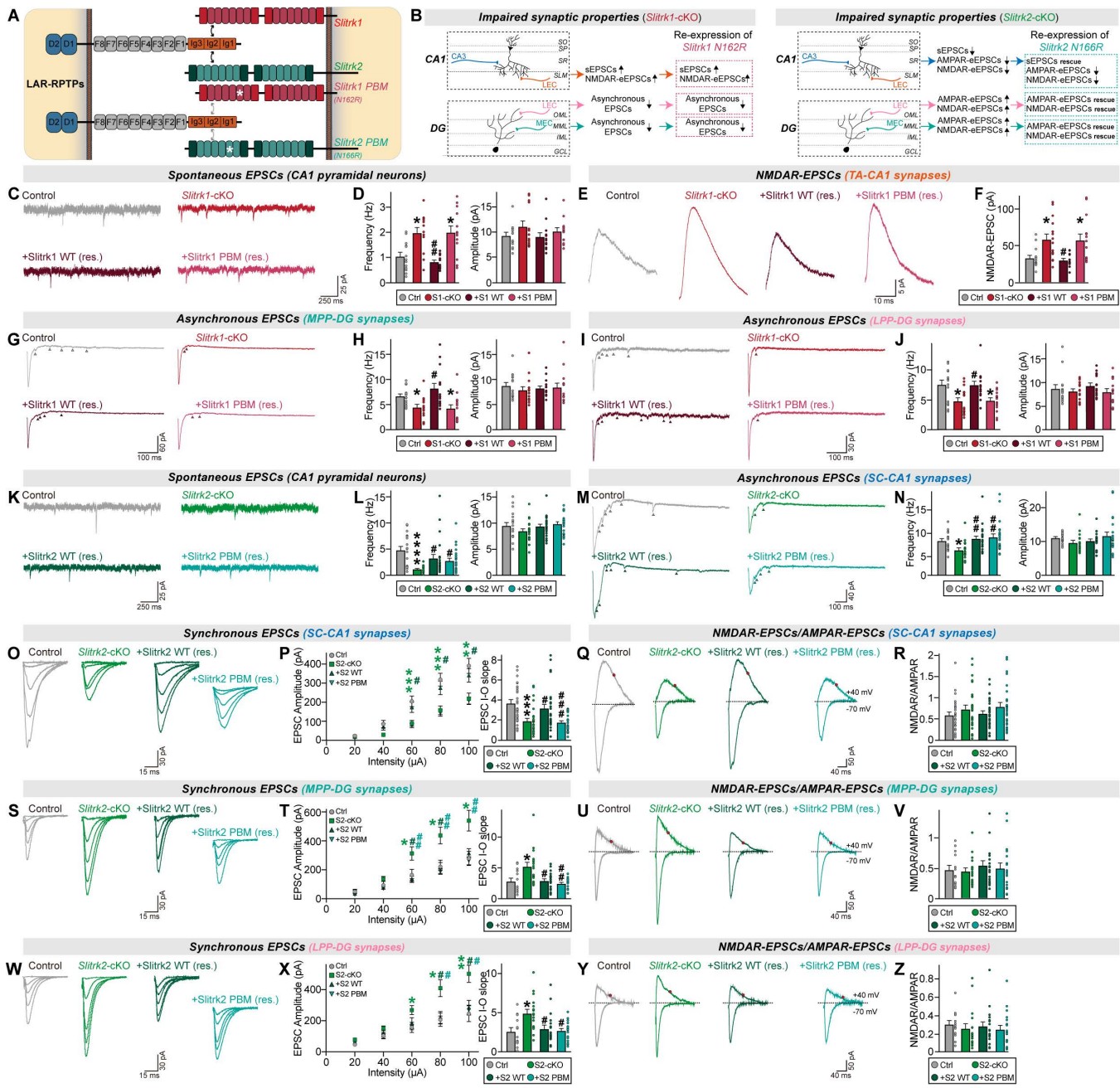

**Fig 5. LAR-RPTP-binding activity of Slitrk1 and Slitrk2 is differentially required for controlling distinct excitatory synaptic properties in a circuit context-dependent manner. (A)** Schematic showing binding mutants of *Slitrk1* N162R (Slitrk1 PBM) and *Slitrk2* N166R (Slitrk2 PBM) to LAR-RPTPs. Abbreviation: PBM, LAR-RPTP-binding mutant. **(B)** Summary of impaired synaptic properties by *Slitrk1*-cKO (left) and *Slitrk2*-cKO (right) mice in CA1 and DG. **(C and D)** Whole-cell recordings of sEPSCs in CA1 pyramidal neurons from control, *Slitrk1*-cKO, or *Slitrk1*-cKO expressing Slitrk1 WT or Slitrk1 PBM. Representative traces (C) and averages of sEPSCs (D; control, *n* = 13/4; *Slitrk1*-cKO, *n* = 13/4; +Slitrk1 WT, *n* = 13/4; +Slitrk1 PBM, *n* = 13/4, where '*n*' denotes the number of cells/mice). **(E and F)** Same as (C and D), except that NMDAR-EPSCs were measured at TA-CA1 synapses. Representative traces (E) and averages of NMDAR-EPSCs (F; control, *n* = 15/5; *Slitrk1*-cKO, *n* = 15/5; +Slitrk1 WT, *n* = 13/5; +Slitrk1 PBM, *n* = 14/5). **(G and H)** Same as (C and D), except that aEPSCs were measured at MPP-DG synapses. Representative traces (G) and averages of aEPSCs (H; control, *n* = 15/5; *Slitrk1*-cKO, *n* = 15/5; +Slitrk1 WT, *n* = 15/5; +Slitrk1 PBM, *n* = 15/5). **(I and J)** Same as (C and D), except that aEPSCs were measured at LPP-DG synapses. Representative traces (I) and averages of aEPSCs (J; control, *n* = 15/5; *Slitrk1*-cKO, *n* = 15/5; +Slitrk1 WT, *n* = 15/5; +Slitrk1 PBM, *n* = 15/5). **(K and L)** Whole-cell recordings of sEPSCs from CA1 pyramidal neurons of control, *Slitrk2*-cKO, or *Slitrk2*-cKO expressing Slitrk2 WT or Slitrk2 PBM. Representative traces (K) and averages of sEPSCs (L; control, *n* = 19/7; *Slitrk2*-cKO, *n* = 21/7; +Slitrk2 WT, *n* = 23/7; +Slitrk2 PBM, *n* = 23/7). **(M**

and **N)** Same as (K and L), except that aEPSCs were measured at SC-CA1 synapses. Representative traces (M) and averages of aEPSCs (N; control, $n$ = 16/7; *Slitrk2*-cKO, $n$ = 17/7; +Slitrk2 WT, $n$ = 24/7; +Slitrk2 PBM, $n$ = 22/7). **(O and P)** Same as (K and L), except that eEPSCs were measured at SC-CA1 synapses. Representative traces (O) and averages of eEPSCs (P; control, $n$ = 27/8; *Slitrk2*-cKO, $n$ = 28/8; +Slitrk2 WT, $n$ = 30/8; +Slitrk2 PBM, $n$ = 26/8). **(Q and R)** Same as (K and L), except that AMPAR/NMDAR-EPSCs were measured at SC-CA1 synapses. Representative traces (Q) and averages of NMDAR/AMPAR-EPSCs (R; control, $n$ = 27/8; *Slitrk2*-cKO, $n$ = 28/8; +Slitrk2 WT, $n$ = 30/8; +Slitrk2 PBM, $n$ = 26/8). **(S and T)** Same as (K and L), except that eEPSCs were measured at MPP-DG synapses. Representative traces (S) and averages of eEPSCs (T; control, $n$ = 13/4; *Slitrk2*-cKO, $n$ = 18/5; +Slitrk2 WT, $n$ = 19/5; +Slitrk2 PBM, $n$ = 18/5). **(U and V)** Same as (K and L), except that NMDAR/AMPAR-EPSCs were measured at MPP-DG synapses. Representative traces (U) and averages of NMDAR/AMPAR-EPSCs (V; control, $n$ = 13/4; *Slitrk2*-cKO, $n$ = 17/5; +Slitrk2 WT, $n$ = 19/5; +Slitrk2 PBM, $n$ = 18/5). **(W and X)** Same as (K and L), except that eEPSCs were measured at LPP-DG synapses. Representative traces (W) and averages of eEPSCs (X; control, $n$ = 13/4; *Slitrk2*-cKO, $n$ = 17/5; +Slitrk2 WT, $n$ = 18/5; +Slitrk2 PBM, $n$ = 18/5). **(Y and Z)** Same as (K and L), except that NMDAR/AMPAR-EPSCs were measured at LPP-DG synapses. Representative traces (Y) and averages of NMDAR/AMPAR-EPSCs (Z; control, $n$ = 13/4; *Slitrk2*-cKO, $n$ = 17/5; +Slitrk2 WT, $n$ = 18/5; +Slitrk2 PBM, $n$ = 18/5). Data are presented as means ± SEMs (*$p$ < 0.05, ***$p$ < 0.001, ****$p$ < 0.0001, #$p$ < 0.05, ##$p$ < 0.01, ###$p$ < 0.001; '#' indicates statistical comparisons of Slitrk1 PBM or Slitrk2 PBM with their counterparts; Kruskal–Wallis test followed by Dunn's multiple comparison test). Numerical data can be found in S1 Data.

expressing Slitrk2 ΔPDZ or Slitrk2 WT, or AAV-ΔCre or AAV-Cre alone (controls), and recorded sEPSCs in CA1 pyramidal neurons, eEPSCs and aEPSCs at SC-CA1 synapses, and eEPSCs at MPP-DG and LPP-DG synapses (Fig 6). Expression of Slitrk2 ΔPDZ completely rescued the amplitude of eEPSCs and frequency of aEPSCs at SC-CA1 synapses, but failed to rescue the frequency of sEPSCs (Fig 6C–6J). These results suggest that Slitrk2 interactions with PDZ proteins are required for maintenance of spontaneous synaptic transmission at Schaffer collateral circuits, but not for regulating synchronous or asynchronous synaptic transmission at these circuits (Fig 6B). Intriguingly, the expression of Slitrk2 ΔPDZ was also able to rescue the abnormalities in synchronous synaptic transmission at both perforant pathways in DG granule neurons (Fig 6K–6R).

Because there are currently no TrkB binding-deficient Slitrk2 variants available that would allow direct testing of the dependence of Slitrk2 activity on TrkB binding, the effects of postsynaptic TrkB deletion on synaptic properties were analyzed using mice with cKO of the *Ntrk2* gene encoding TrkB; these results were validated by semi-quantitative immunoblotting analyses using CA1 lysates from adult *Ntrk2*-cKO (S14A and S14B Fig). We found that postsynaptic deletion of TrkB in CA1 neurons of these *Ntrk2*-cKO mice significantly reduced the amplitude of eEPSCs and the frequency of aEPSCs at SC-CA1 synapses without altering excitability and sEPSCs in CA1 pyramidal neurons (S4M, S4N, and S14C–S14K Figs). In contrast, similar ablation of postsynaptic TrkB in DG neurons did not alter eEPSC amplitudes at either MPP-DG or LPP-DG synapses (S14L–S14U Fig). Taken together, these results suggest that Slitrk2 governs distinct excitatory synaptic properties through distinct extracellular and intracellular mechanisms in a microcircuit-context-dependent fashion (S4 Table).

## Pathogenic Slitrk2V89M substitution impairs subsets of Slitrk2 synaptic functions and spatial reference memory in adult mice

New reports continue to link a variety of *SLITRK2* mutations to diverse neurodevelopmental, neuropsychiatric, and neurodegenerative disorders [17,51,52]. Most *SLITRK2* mutations appear to be loss-of-function mutations, often affecting the surface expression and trafficking of Slitrk2 in neurons and thereby compromising excitatory synaptic transmission [17,19]. Notably, a valine-to-methionine mutation at residue 89 in *SLITRK2* (V89M) identified in a subset of schizophrenic patients was previously shown to impair excitatory synapse-formation in cultured hippocampal neurons, but without affecting the surface expression, LAR-RPTP-binding activity, or synaptogenic activity of Slitrk2 [19] (S15A and S15B Fig). Thus, we asked whether the pathogenicity of Slitrk2 V89M is related to the impairments in excitatory synaptic properties observed in *Slitrk2*-cKO mice. To test this hypothesis, we introduced the V89M substitution into the endogenous *Slitrk2* gene in mice by clustered regularly interspaced short palindromic repeats (CRISPR)/CRISPR-associated protein 9 (Cas9) technology, to generate Slitrk2 conditional knock-in (*Slitrk2*V89M-cKI) mice, allowing Cre recombinase-dependent Slitrk2 substitution (S15C and S15D Fig). We then administered AAV-Cre into the hippocampal CA1 of *Slitrk2*-V89Mf/f mice and conducted electrophysiological analyses. Sanger sequencing results of the injected tissue region confirmed correct Cre

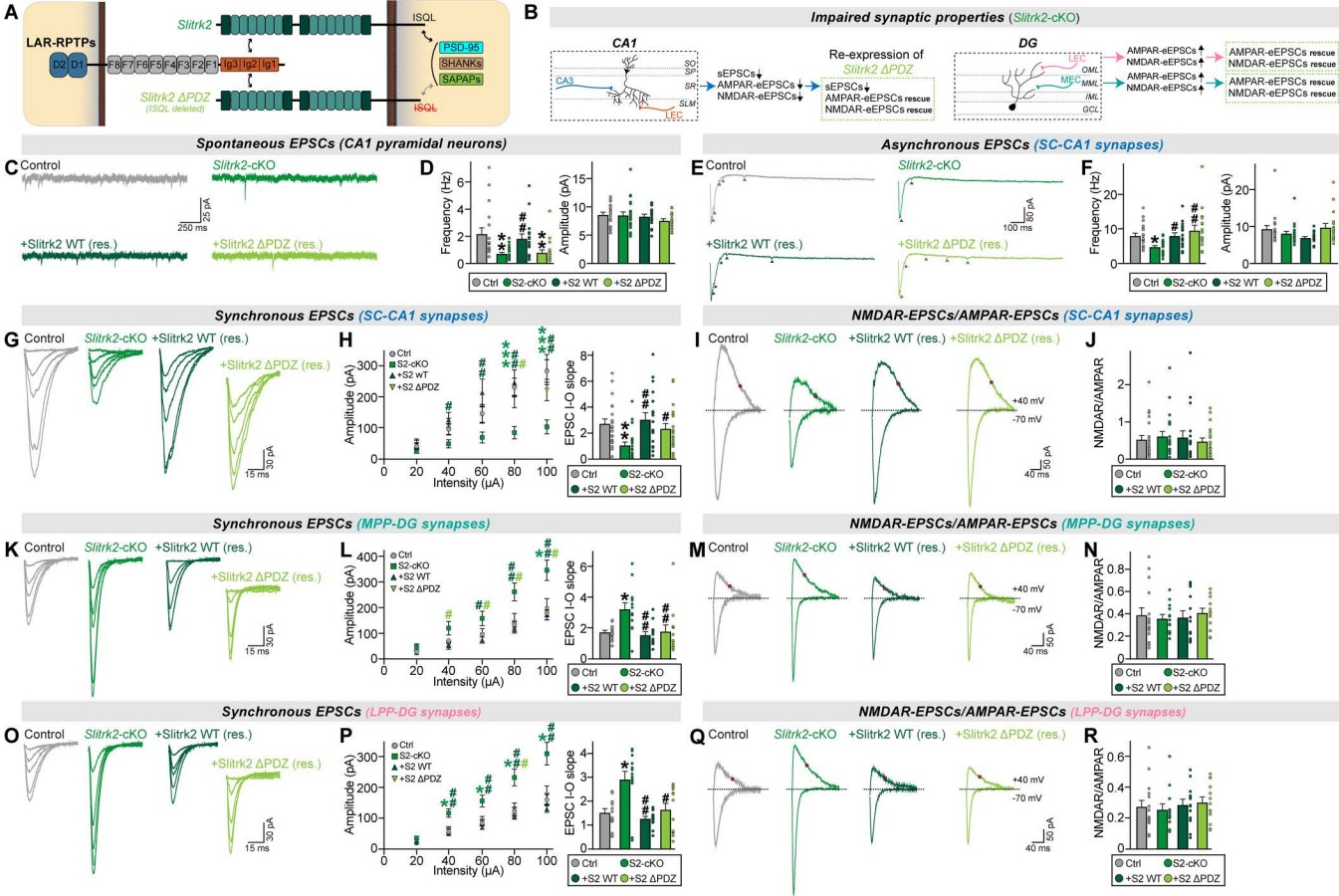

**Fig 6. Slitrk2 binding to PDZ domain-containing proteins is required for its regulation of specific excitatory synapse properties in hippocampal circuits. (A)** Schematic depicting the design of SLITRK2 ΔPDZ, from which the PDZ domain-binding sequence (ISQL) of SLITRK2 has been deleted. **(B)** Summary of impaired synaptic properties by *Slitrk2*-cKO and re-expression effects of SLITRK2 ΔPDZ in CA1 and DG. **(C and D)** Whole-cell recordings of sEPSCs from CA1 pyramidal neurons of control, *Slitrk2*-cKO, *Slitrk2*-cKO expressing Slitrk2 WT, or Slitrk2 ΔPDZ. Representative traces (C) and averages of sEPSCs (D; control, n = 19/5; *Slitrk2*-cKO, n = 21/5; +Slitrk2 WT, n = 23/4; +Slitrk2 ΔPDZ, n = 23/5, where n denotes the number of cells/mice). **(E and F)** Same as (C and D), except that whole-cell recordings of aEPSCs at SC-CA1 synapses were performed. Representative traces (E) and averages of aEPSCs (F; control, n = 16/5; *Slitrk2*-cKO, n = 17/5; +Slitrk2 WT, n = 24/5; +Slitrk2 ΔPDZ, n =22/5). **(G and H)** Same as (C and D), except that whole-cell recordings of eEPSCs at SC-CA1 synapses were performed. Representative traces (G) and averages of eEPSCs (H; control, n = 27/5; *Slitrk2*-cKO, n = 28/5; +Slitrk2 WT, n = 30/5; +Slitrk2 ΔPDZ, n=26/5). **(I and J)** Same as (C and D), except that NMDAR/AMPAR-EPSCs at SC-CA1 synapses were recorded. Representative traces (I) and averages of NMDAR/AMPAR-EPSCs (J; control, n = 27/5; *Slitrk2*-cKO, n = 28/5; +Slitrk2 WT, n = 30/5; +Slitrk2 ΔPDZ, n = 26/5). **(K and L)** Same as (G and H), except that whole-cell recordings of eEPSCs at MPP-DG synapses were performed. Representative traces (K) and averages of eEPSCs (L; control, n = 13/4; *Slitrk2*-cKO, n = 17/4; +Slitrk2 WT, n = 19/4; +Slitrk2 ΔPDZ, n = 18/4). **(M and N)** Same as (I and J), except that NMDAR/AMPAR-EPSCs at MPP-DG synapses were recorded. Representative traces (M) and averages of NMDAR/AMPAR-EPSCs (N; control, n = 13/4; *Slitrk2*-cKO, n = 17/4; +Slitrk2 WT, n = 19/4; +Slitrk2 ΔPDZ, n = 18/4). **(O and P)** Same as (G and H), except that whole-cell recordings of eEPSCs at LPP-DG synapses were performed. Representative traces (O) and averages of eEPSCs (P; control, n = 13/4; *Slitrk2*-cKO, n = 17/4; +Slitrk2 WT, n = 18/4; +Slitrk2 ΔPDZ, n = 18/4). **(Q and R)** Same as (I and J), except that NMDAR/AMPAR-EPSCs at LPP-DG synapses were recorded. Representative traces (Q) and averages of NMDAR/AMPAR-EPSCs (R; control, n = 13/4; *Slitrk2*-cKO, n = 17/4; +Slitrk2 WT, n = 18/4; +Slitrk2 ΔPDZ, n = 18/4). Data are presented as means ± SEMs (*$p < 0.05$, **$p < 0.01$, ***$p < 0.001$, #$p < 0.05$, ##$p < 0.01$; # represents the statistical comparison of Slitrk2 ΔPDZ with Slitrk2 WT; Kruskal–Wallis test followed by Dunn's multiple comparison test). Numerical data can be found in S1 Data.

expression and conversion of the endogenous *Slitrk2* gene product to Slitrk2 V89M. There were no alterations in the expression of various synaptic proteins, including Slitrk2, in the CA1 of *Slitrk2*$^{V89M}$-cKI mice (see S1E and S1F Fig). In addition, *Slitrk2*$^{V89M}$-cKI did not influence the mRNA levels of any Slitrk paralog (S16 Fig).

We first monitored the intrinsic excitability of CA1 pyramidal neurons and found that AP firing was decreased in *Slitrk2*$^{V89M}$-cKI CA1 pyramidal neurons (S4O and S4P Fig). We further analyzed five properties—sEPSCs in CA1 pyramidal neurons, eEPSCs at SC-CA1 synapses, aEPSCs at SC-CA1 synapses, eEPSCs at MPP-DG synapses, and eEPSCs at LPP-DG synapses—in the hippocampal circuits of *Slitrk2*$^{V89M}$-cKI mice (Fig 7A). *Slitrk2*$^{V89M}$ substitution significantly decreased the frequencies of sEPSCs and aEPSCs in CA1 pyramidal neurons, changes similar to those in *Slitrk2*-cKO mice (Fig 7B–7E). However, in contrast to *Slitrk2*-cKO mice, eEPSCs at SC-CA1, MPP-DG, or LPP-DG synapses were unaffected in *Slitrk2*$^{V89M}$-cKI mice (Fig 7F–7T), suggesting that Slitrk2 V89M impairs a subset of excitatory synaptic properties that are altered by a Slitrk2 deficiency (summarized in S4 Table).

We previously showed that hippocampal CA1-specific deletion of Slitrk2 impairs spatial reference memory; this was assessed by the Barnes maze test and represented the sole identified behavioral deficit associated with the CA1-specific cKO of Slitrk2 [17]. Here, we set out to directly test the hypothesis that the pathogenic *Slitrk2* V89M will recapitulate this known CA1-dependent behavioral phenotype. We thus examined whether mice with a hippocampal CA1-specific KI of *Slitrk2* V89M exhibited similar deficits in the Barnes maze test (Fig 7U–7Z). Consistent with the previously reported behavioral phenotypes of *Slitrk2*-cKO mice, CA1-specific *Slitrk2*$^{V89M}$-cKI mice exhibited long-term spatial memory deficits, such as impaired retention of spatial reference memory (Fig 7V–7Z). Taken together, these electrophysiological and behavioral analyses underscore the significance of circuit-specific impairments in pinpointing plausible pathophysiological mechanism(s) underlying brain disorders.

## Discussion

Mounting evidence indicates that synaptic CAMs are critical for shaping key synaptic properties. Although this view has motivated an enormous increase in descriptive studies of various synaptic CAMs, our mechanistic understanding has lagged. The nature of the interplay between various synaptic CAMs, particularly between different paralogs in the same CAM family, has been largely unexplored. In the present study, we addressed the circuit-specific roles of two excitatory Slitrk paralogs, Slitrk1 and Slitrk2, in specifying postsynaptic properties, including spontaneous, synchronous, and asynchronous synaptic transmission, in two distinct hippocampal subfields, CA1 and DG. We chose these hippocampal subfields because these two regions are anatomically well segregated, and the detailed microcircuit wiring is well established. We used conditional KO mice to specifically probe the postsynaptic role of these proteins and monitored whether single or double genetic deletion of these Slitrk paralogs influences different modes of synaptic transmission/strength in a defined hippocampal microcircuit. We obtained four principal findings.

First, Slitrk1 and Slitrk2 exert opposite effects on spontaneous synaptic transmission and excitability (intrinsic property) in the hippocampal CA1 of adult mice. The effect of *Slitrk1*-cKO on sEPSCs was identical to that previously reported in *Slitrk1*-KD rats [18] (see S1 Table). However, there were striking differences in the effect of different Slitrk1 loss-of-function genetic manipulations. For example, instead of changes in the frequency of aEPSCs at SC-CA1 synapses or amplitude of AMPAR-EPSCs, *Slitrk1*-cKO mice exhibited increased amplitudes of NMDAR-EPSCs at TA-CA1 synapses. Several factors could contribute to these discrepancies, including differences in the extent of gene suppression (~50% for KD versus ~90% for cKO), potential off-target effects of KD neurons, differences in the techniques used for genetic manipulations, and differences in model species (rat versus mouse) and model ages (P13–P19 versus 6–7 weeks). Notwithstanding these differences, our study reinforces the idea that cKO analyses are essential for reaffirming the circuit-specific role of a synaptic CAM, and that an individual protein might perform distinct roles at specific developmental stage.

Second, Slitrk1 and Slitrk2 are not functionally redundant in regulating synaptic properties, at least in hippocampal excitatory circuits. Simultaneous deletion of Slitrk1 and Slitrk2 in CA1 pyramidal neurons canceled out the dichotomous

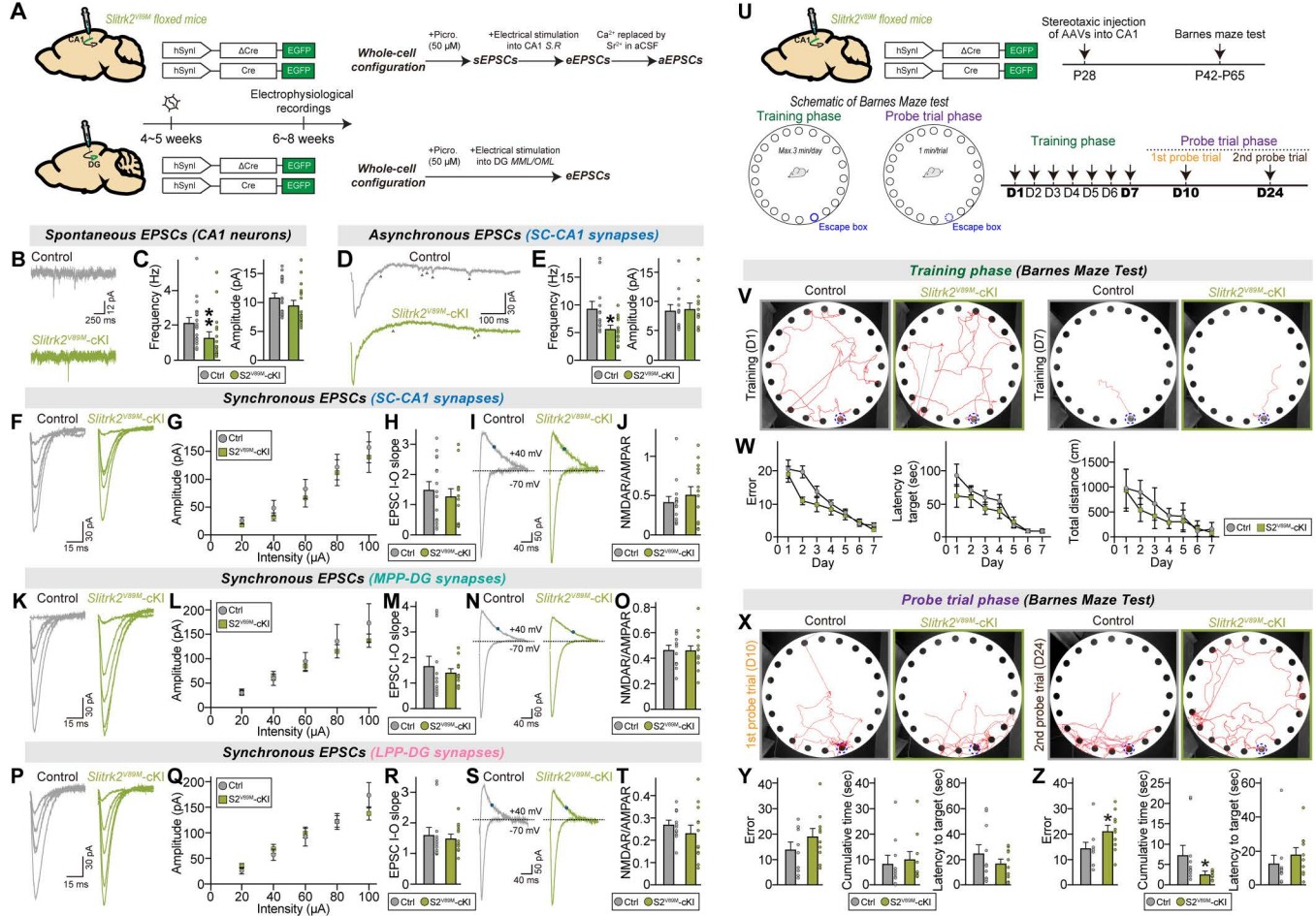

**Fig 7. Pathogenic Slitrk2 V89M impairs basal and asynchronous excitatory synaptic transmission in hippocampal CA1 circuits and spatial reference memory. (A)** Schematic showing injections of the indicated AAVs into the hippocampal CA1 or DG of *Slitrk2*-V89M^f/f mice. Electrophysiological recordings were performed 2 weeks after injections to monitor alterations in spontaneous, synchronous and asynchronous synaptic transmission. **(B and C)** Whole-cell recordings of sEPSCs from CA1 pyramidal neurons of control and *Slitrk2*^V89M-cKI hippocampal slices. Representative traces (B) and averages of sEPSCs (C; control, *n* = 19/5; *Slitrk2*^V89M-cKI, *n* = 17/5, where '*n*' denotes the number of cells/mice). **(D and E)** Whole-cell recordings of aEPSCs at SC-CA1 synapses of control and *Slitrk2*^V89M-cKI hippocampal slices. Representative traces (D) and averages of aEPSCs (E; control, *n* = 12/4; *Slitrk2*^V89M-cKI, *n* = 13/4). **(F–J)** Whole-cell recordings of eEPSCs at SC-CA1 synapses of control and *Slitrk2*^V89M-cKI hippocampal slices. Representative traces (F) and averages of eEPSCs (G and H; control, *n* = 15/4; *Slitrk2*^V89M-cKI, *n* = 14/4, where '*n*' denotes the number of cells/mice). Representative traces of NMDAR/AMPAR-EPSC at SC-CA1 synapses of control and *Slitrk2*^V89M-cKI hippocampal slice (I) and averages of NMDAR/AMPAR-EPSCs (J; *n* = 15/4; *Slitrk2*^V89M-cKI, *n* = 14/4). **(K–O)** Whole-cell recordings of eEPSCs at MPP-DG synapses of control and *Slitrk2*^V89M-cKI hippocampal slices. Representative traces (K) and averages of eEPSCs (L and M; control, *n* = 12/4; *Slitrk2*^V89M-cKI, *n* = 12/4). Representative traces of NMDAR/AMPAR-EPSC at MPP-DG synapses of control and *Slitrk2*^V89M-cKI hippocampal slice (N) and averages of NMDAR/AMPAR-EPSCs (O; control, *n* = 12/4; *Slitrk2*^V89M-cKI, *n* = 12/4). **(P–T)** Whole-cell recordings of eEPSCs at LPP-DG synapses of control and *Slitrk2*^V89M-cKI hippocampal slice. Representative traces (P) and averages of eEPSCs (Q and R; control, *n* = 12/4; *Slitrk2*^V89M-cKI, *n* = 12/4). Representative traces of NMDAR/AMPAR-EPSCs at LPP-DG synapses of Control and *Slitrk2*^V89M-cKI hippocampal slices (S) and averages of NMDAR/AMPAR-EPSCs (T; *n* = 12/4; *Slitrk2*^V89M-cKI, *n* = 12/4). Data are presented as means ± SEMs (*$p < 0.05$, **$p < 0.01$; two-tailed nonparametric Mann–Whitney $U$ test). **(U)** Schematic of the Barnes maze test. **(V and W)** Representative tracking images (V) of control and *Slitrk2*^V89M-cKI mice on the first (day 1) and last (day 7) days of training. Number of errors before first encountering the escape hole (W; left), escape latency (W; middle) and total distance (W; right) for control and *Slitrk2*^V89M-cKI mice during the training session. Data are presented as means ± SEMs (control, *n* = 10; *Slitrk2*^V89M-cKI, *n* = 10). **(X–Z)** Representative tracking images (X) of control and *Slitrk2*^V89M-cKI mice on the first (day 10) and second (day 24) days of the probe trials. Number of errors before first encountering the escape hole for control and *Slitrk2*^V89M-cKI mice during the first (Y) and second probe (Z) trials. Representative track images (bottom) from the first and second probe trials. Data are presented as means ± SEMs (control, *n* = 10; *Slitrk2*^V89M-cKI, *n* = 10; *$p < 0.05$, **$p < 0.01$; two-tailed nonparametric Mann–Whitney $U$ test). Numerical data can be found in S1 Data.

action of each Slitrk paralog on excitability and spontaneous synaptic transmission, suggesting that both Slitrk proteins operate through a converging pathway. However, Slitrk1 and Slitrk2 function in a nonredundant manner to regulate other synaptic properties, such that each Slitrk paralog is required for proper maintenance. The nonredundant actions of each Slitrk protein do not appear to be attributable to their localization patterns within the hippocampal subfields, as they are widely expressed in various layers of the hippocampal CA1 and DG, unlike the unique distribution patterns of other synaptic CAMs (e.g., NGL-1/Netrin-G1, NGL-2/Netrin-G2, or Lphn2/Lphn3) [5,53]. Thus, we propose that Slitrk1 and Slitrk2 likely operate with other nonfamily members of synaptic CAMs to shape the diversity of synaptic properties of the respective excitatory neurons (pyramidal and granule neurons) in the hippocampus. From this perspective, Slitrks might be functionally analogous to Nlgns, in that both CAMs are involved in the regulation of excitatory and inhibitory synapses in a paralog-dependent manner and their respective expression is not restricted to specific neurons of specific layers within the hippocampus. As emphasized in most prior literature reviews [e.g., 2,12,32], Slitrk4 and Slitrk5 may be similarly required for the maintenance of excitatory synapses in cultured hippocampal neurons. This notion was also suggested in our previous study [10], although Slitrk5 has also been reported to negatively regulate inhibitory synapses in dopaminergic neurons [54]. The current study, however, dispelled the concept. Future studies should seek to determine whether Slitrk4 and Slitrk5, like Slitrk1 and Slitrk2, have different functions in the context of a defined circuit. This information would assist with developing a comprehensive understanding of the role of Slitrk family proteins in organizing synaptic properties.

Third, consistent with the unique action of Slitrk1 and Slitrk2 in governing synaptic properties, both proteins control distinct postsynaptic properties through different mechanisms. Slitrk1 requires exclusively its ability to bind LAR-RPTPs to regulate excitatory synaptic transmission and strength in both CA1 and DG neurons. Unlike Slitrk1, Slitrk2 binds to TrkB receptor tyrosine kinase and PDZ domain-containing scaffolds in addition to LAR-RPTPs; our *Ntrk2*-cKO analyses and molecular replacement experiments demonstrated that these unique molecular interactions of Slitrk2 contribute to the individual shaping of specific synaptic properties in hippocampal SC-CA1 circuits. Notably, however, the ability of Slitrk2 to bind LAR-RPTPs, PDZ proteins or TrkB is not required for Slitrk2 to regulate synaptic properties within DG granule neurons, suggesting the existence of additional, as yet unidentified, binding partners for Slitrk2. Deploying different *cis*- and *trans*-cellular ligands for shaping specific synaptic properties in distinct neural circuits has not been systematically tested and/or documented for other synaptic CAMs; however, our present results are reminiscent of those of our previous finding that presynaptic PTPδ regulates NMDAR-mediated postsynaptic responses in a circuit context-dependent manner [48]. Interestingly, Slitrk1 intracellularly interacts with 14-3-3 proteins to orchestrate various phosphorylation cascades in neurons [55]. Because 14-3-3 proteins are involved in positively regulating NMDA receptors [56,57], it is tempting to speculate that, although Slitrk1-mediated intracellular mechanisms are not necessary for the maintenance of proper synaptic properties, the interaction of Slitrk1 with 14-3-3 might also contribute to the regulation of NMDARs at TA-CA1 synapses. Future studies are warranted to directly address this possibility, ideally using cKO mice in which all 14-3-3 isoforms are deleted [58].

Fourth, the Slitrk2$^{V89M}$ mutation recapitulates the effects of *Slitrk2*-cKO on spontaneous and asynchronous, but not synchronous, synaptic transmission in hippocampal circuits. These results suggest that Slitrk2$^{V89M}$ acts as a loss-of-function mutation in the hippocampal microcircuit, where postsynaptic loss of Slitrk2 yields profound deficits in excitatory synaptic properties. Our previous study showed that Slitrk2$^{V89M}$ eliminates the ability of Slitrk2 WT to facilitate excitatory synapse density without affecting its ability to promote presynaptic assembly [19]. Thus, it was unclear how this mutant protein exerts pathogenic effects in neurons. The current study demonstrated that circuit-specific impairment of a particular synaptic transmission mode might underlie the manifestations of brain disorders associated with Slitrk2 dysregulation. Indeed, CA1-specific *Slitrk2*$^{V89M}$-cKI mice exhibited impaired spatial reference memory, recapitulating the behavioral deficits of CA1-specific *Slitrk2*-cKO mice [17]. The Slitrk2 T312A mutation identified in patients with neurodevelopmental disorders appears to exert phenotypes similar to Slitrk2 V89M [17]; thus, it would be extremely interesting to use *Slitrk2*$^{T312A}$-cKI mice to examine whether this mutation yields similar or distinct alterations in the synaptic properties identified in the current study. Future studies targeting other cell types, circuits, and/or brain regions would provide more

integrated insights into how Slitrk dysfunctions are linked to the pathogenic mechanisms underlying associated brain disorders. We also note that there appears to be a discrepancy between the profound reduction in morphological synapses and the comparatively modest decrease in mEPSC frequency. However, we propose that this is not a technical contradiction but rather a key biological implication. Two nonmutually exclusive explanations are plausible: (i) the two assays may sample distinct synaptic populations (e.g., mEPSCs from CA1 neurons versus layer-specific imaging of all synaptic puncta), and/or (ii) potent homeostatic compensatory mechanisms, such as an increased presynaptic release probability at remaining synapses, may functionally offset the structural loss. Clarifying these complex adaptations remains an important task for future studies.

The current landscape of transsynaptic pathways suggests that a few presynaptic CAMs bind to multifarious postsynaptic CAMs [1], suggesting that each postsynaptic CAM might instruct the diversity and complexity of neural circuits. However, only a few postsynaptic CAMs are important for shaping synaptic specificity [5,18,29,59]. The current study demonstrated that Slitrk1 and Slitrk2 also act as *bona fide* synaptic CAMs in dictating specific synaptic properties *in vivo*. However, given that other synaptic CAMs are likely to play similar roles in the same microcircuit, it is crucial to determine whether and how Slitrks combine with other (specific) CAMs to govern specific synaptic properties. LRRTMs and Nlgns function together in specifying AMPAR functions in hippocampal CA1 pyramidal neurons by virtue of their abilities to bind presynaptic Nrxns [44,28,60]. However, LRRTM1 and SynCAM1, which do not share any binding proteins, also operate together in organizing excitatory synapses in the prefrontal cortex [61]. Thus, it is difficult to predict the combinatorial pair(s) of synaptic CAMs that play similar roles based on protein-protein interaction maps and/or expression patterns of a specific CAM. From this perspective, the current study paints a perplexing picture by demonstrating that even structurally similar paralogs within a single CAM family perform different functions through disparate mechanisms (see S17 Fig). Although we were unable to detect differences in the synaptic localization of Slitrk1 and Slitrk2 at nanoscale resolution, it is plausible that both Slitrks localize to discrete subsynaptic domains at excitatory synapses [62]. Another important question that needs to be addressed is how the binding of Slitrks to the same ligands (i.e., LAR-RPTPs) within the same microcircuit leads to different (often opposite) outcomes. Although both Slitrk1 and Slitrk2 activate LAR-RPTP-mediated presynaptic assembly in neurons [23], the transsynaptic signals produced by distinct Slitrk paralogs may be differentially processed onto presynaptic neurons *in vivo*. Consistent with this conjecture, the deletion of each Slitrk paralog resulted in distinct alterations in presynaptic vesicle organization. Future studies encompassing combined loss-of-function approaches targeting various cell types, circuit types, and brain regions would help assemble the molecular logic underlying how synaptic specificity is encoded by synaptic CAMs.

## Supporting information

**S1 Fig. Validation of Slitrk1 and Slitrk2 antibodies used in the current study and analyses of the effects of conditional deletion of Slitrk proteins on synaptic protein expression. (A)** Strategy used to generate *Slitrk1*-cKO mice. LoxP sites were introduced at positions flanking the neomycin gene, FLP recombinant target (FRT), and exon 1 (E1) of the murine *Slitrk1* gene. Black arrows indicate forward and reverse primers used for genotyping. Note that lacZ and neomycin cassettes are two separate markers. **(B)** PCR genotyping of *Slitrk1*-floxed mice. The band size of the *Slitrk1*-floxed allele was 368 bp. **(C)** Quantitative RT-PCR analyses of *Slitrk* mRNA levels in the hippocampal lysates of *Slitrk1*-cKO mice. Data are presented as means ± SEMs ($n = 3$ mice/group). **(D)** Immunohistochemical validation of commercially available anti-Slitrk1 and anti-Slitrk2 antibodies using brain slices from *Slitrk1*-cKO (left) and *Slitrk2*-cKO (right) mice, respectively. The indicated *Slitrk* floxed mice were stereotactically injected with AAVs expressing Cre recombinase or inactive Cre (ΔCre; control), with immunohistochemical analyses performed 2 weeks later. Scale bar, 50 μm (applies to all images). Abbreviations: SO, *stratum oriens*; SP, *stratum pyramidale*; SR, *stratum radiatum*. **(E and F)** Representative immunoblots (E) of hippocampal lysates from *Slitrk1*<sup>f/f</sup>, *Slitrk2*<sup>f/f</sup>, *Slitrk2* V89M<sup>f/f</sup>, and *Slitrk1/2*<sup>f/f</sup> mice expressing Cre or ΔCre

(control). Summary graphs (F) showing levels of various synaptic proteins, analyzed by semi-quantitative immunoblotting. Data are presented as means±SEMs ($n=4$–8 mice/group; $*p<0.05$, $**p<0.01$, $***p<0.001$; two-tailed unpaired $t$ test). A red asterisk indicates nonspecific bands. Numerical data can be found in S1 Data. **S2 Fig. Validation of Slitrk1 and Slitrk2 antibodies used in eMAP tissues derived from *Slitrk*-cKO mice. (A** and **B)** Immunohistochemical staining-based validation of the antibodies recognizing Slitrk1 (top) and Slitrk2 (bottom) with the postsynaptic markers, SHANK2 and PSD-95, in the SR (A) and SLM (B) regions of the CA1 hippocampus of ΔCre control and Cre cKO mice. *Slitrk*-floxed mice were stereotactically injected with AAVs expressing Cre recombinase or inactive Cre (ΔCre; control), and immunohistochemical analyses were performed 3 weeks later. Scale bars=2 μm (applies to all images). Abbreviations: SR, *stratum radiatum*; SLM, *stratum lacunosum-moleculare*. **(C** and **D)** Quantitative analysis of (C) the total number of Slitrk1 puncta for ΔCre control (Ctrl) and Cre cKO mice in the SR and SLM layers of the CA1 hippocampus and (D) the total number of PSD-95 and SHANK2 synaptic puncta from the corresponding SR and SLM layer images analyzed in (A and B). Data are presented as means±SEMs ($n=3$ mice/group; $*p<0.05$; one-tailed Mann–Whitney $U$ test). **(E** and **F)** Quantitative analysis of (E) the total number of Slitrk2 puncta for ΔCre control (Ctrl) and Cre cKO mice in the SR and SLM layers of the CA1 hippocampus and (F) the total number of PSD-95 and SHANK2 synaptic puncta from the corresponding SR and SLM layer images analyzed in (A and B). Data are presented as means±SEMs ($n=3$ mice/group). **(G** and **H)** Immunohistochemical staining-based validation of antibodies against Slitrk1 (top) and Slitrk2 (bottom) with postsynaptic markers, SHANK2 and PSD-95 in the OML (G) and MML (H) regions of the DG of ΔCre control and Cre cKO mice. Scale bars=2 μm (applies to all images). Abbreviations: OML, outer molecular layer; MML, middle molecular layer. **(I** and **J)** The same quantitative analysis of Slitrk1, PSD-95, and SHANK2 puncta from (C and D) but in the OML and MML layers of the DG. Data are presented as means±SEMs ($n=3$ mice/group; $*p<0.05$; one-tailed Mann–Whitney $U$ test). **(K** and **L)** The same quantitative analysis of Slitrk1, PSD-95, and SHANK2 puncta from (E and F) but in the OML and MML layers of the DG. Puncta were measured from volumetric images from three separate gels (3 images/ gel) for each mouse, and values were averaged for each mouse. Data are presented as means±SEMs ($n=3$ mice/group). Numerical data can be found in S1 Data. **S3 Fig. Immunohistochemical analyses showing the virus targeting and virus transduction efficiency in hippocampal CA1 and DG from adult *Slitrk*-cKO mice. (A)** Representative images showing that stereotactic injections reliably targeted AAVs to the CA1 and dentate gyrus (DG) in adult male *Slitrk*-cKO mice. GFP fluorescence indicates virus-transduced neurons. Scale bar: 200 μm. **(B)** Immunohistochemical analyses of viral transduction efficiency. Sections were stained with anti-NeuN (to label neurons) and anti-GFP antibodies. Colocalization of GFP with NeuN was used to verify cell-type-specific infection. Scale bar: 50 μm. **(C)** Quantification of transduction efficiency, expressed as the percentage of GFP$^+$/NeuN$^+$ double-positive cells among total NeuN$^+$ neurons. Data are presented as means±SEMs ('$n$' denotes number of images; ΔCre, $n=3$, Cre, $n=3$; two-tailed nonparametric Mann–Whitney $U$ test). Numerical data can be found in S1 Data. **S4 Fig. Measurement of cell excitability in hippocampal CA1 pyramidal neurons and DG granule neurons from *Slitrk1*-cKO, *Slitrk2*-cKO, *Slitrk1/2*-dKO, *Ntrk2*-cKO, and *Slitrk2*$^{V89M}$-cKI mice. (A–D**, **I**, **J**, **M–P)** Whole-cell recordings of action potential (AP) firings from CA1 pyramidal neurons in control (gray), *Slitrk1*-cKO (red), *Slitrk2*-cKO (green), *Slitrk1/2*-dKO (orange), *Ntrk2*-cKO (blue), and *Slitrk2*$^{V89M}$-cKI (light-green) hippocampal slices. Representative traces (A**, C, I**, M and O) and averages of AP firings (B, D, J, N and P; control in (B), $n=12/3$; control in (D), $n=10/4$; control in (J), $n=13/4$; control in (N), $n=14/3$; control in (P), $n=13/3$; *Slitrk1*-cKO, $n=11/4$; *Slitrk2*-cKO, $n=15/4$; *Slitrk1/2*-dKO, $n=11/4$; and *Ntrk2*-cKO, $n=15/3$; and *Slitrk2*$^{V89M}$-cKI, $n=14/3$, where '$n$' denotes number of cells/mice. **(E–H, K, L)** Same as (A–D, I, J, M–P), except that whole-cell recordings of representative AP firings from DG granule neurons were performed. Representative traces (E, G and K) and averages of AP firings (F, H and L; control in (F), $n=12/4$; control in (H), $n=21/5$; control in (L), $n=11/4$; *Slitrk1*-cKO, $n=12/4$; *Slitrk2*-cKO, $n=17/5$; and *Slitrk1/2*-dKO, $n=10/4$). Data are presented as means±SEMs ($*p<0.05$, $**p<0.01$; two-tailed nonparametric Mann–Whitney $U$ test). Numerical data can be found in S1 Data. **S5 Fig. Validation for the ability of DREADD-based chemogenetics to normalize excitabilities of CA1 pyramidal neurons from adult *Slitrk*-cKO mice. (A)** Schematic showing AAV injection into the hippocampal CA1 of *Slitrk1*$^{f/f}$, *Slitrk2*$^{f/f}$ and

*Slitrk2*-V89M$^{f/f}$ mice. Electrophysiological recordings were performed 2 weeks after injections. **(B** and **C)** Whole-cell recordings of action potential (AP) firings from CA1 pyramidal neurons in control (dark-gray), *Slitrk1*-cKO (red), control+hM4Di (light-gray), and *Slitrk1*-cKO+hM4Di (orange) hippocampal slices. Representative traces (B) and averages of AP firings (C; control, *n* = 10/2; *Slitrk1*-cKO, *n* = 9/2; control+hM4Di, *n* = 10/2; *Slitrk1*-cKO+hM4Di, *n* = 10/2, where n denotes the number of cells/mice). **(D** and **E)** Whole-cell recordings of AP firings from CA1 pyramidal neurons in control (dark-gray), *Slitrk2*-cKO (green), control+hM3Dq (light-gray), and *Slitrk2*-cKO+hM3Dq (light-green) hippocampal slices. Representative traces (D) and averages of AP firings (E; control, *n* = 16/2; *Slitrk2*-cKO, *n* = 13/2; control+hM3Dq, *n* = 12/2; *Slitrk2*-cKO+hM3Dq, *n* = 14/2, where *n* denotes the number of cells/mice). **(F** and **G)** Whole-cell recordings of AP firings from CA1 pyramidal neurons in control (dark-gray), *Slitrk2*$^{V89M}$-cKI (lime), control+hM3Dq (light-gray), and *Slitrk2*$^{V89M}$-cKI+hM3Dq (dark-lime) hippocampal slices. Representative traces (F) and averages of AP firings (G; control, *n* = 11/2; *Slitrk2*$^{V89M}$-cKI, *n* = 12/2; control+hM3Dq, *n* = 13/2; *Slitrk2*$^{V89M}$-cKI+hM3Dq, *n* = 14/2, where *n* denotes the number of cells/mice). Data are presented as means ± SEMs (*$p < 0.05$, **$p < 0.01$, #$< 0.05$; '#' indicates statistical comparisons with their counterparts; nonparametric Kruskal–Wallis test with Dunn's *post hoc* test). Numerical data can be found in S1 Data.

**S6 Fig. Normalizing excitability of CA1 pyramidal neurons from adult *Slitrk*-cKO and *Slitrk2*$^{V89M}$-cKI mice does not affect altered spontaneous excitatory synaptic transmission. (A)** Schematic showing AAV injection into the hippocampal CA1 of *Slitrk1*$^{f/f}$, *Slitrk2*$^{f/f}$ and *Slitrk2*-V89M$^{f/f}$ mice. Electrophysiological recordings were performed 2 weeks after injections. **(B** and **C)** Whole-cell recordings of sEPSCs from CA1 pyramidal neurons in control (dark-gray), *Slitrk1*-cKO (red), control+hM4Di (light-gray), *Slitrk1*-cKO+hM4Di (orange) hippocampal slices. Representative traces (B) and averages of sEPSC frequency and amplitude (C; control, n = 9/2; *Slitrk1*-cKO, *n* = 10/2; control+hM4Di, *n* = 8/2; *Slitrk1*-cKO+hM4Di, *n* = 10/2, where n denotes the number of cells/mice). **(D** and **E)** Whole-cell recordings of sEPSCs from CA1 pyramidal neurons in control (dark-gray), *Slitrk2*-cKO (green), control+hM3Dq (light-gray), and *Slitrk2*-cKO+hM3Dq (light-green) hippocampal slices. Representative traces (D) and averages of sEPSC frequency and amplitude (E; control, *n* = 10/2; *Slitrk2*-cKO, *n* = 10/2; control+hM3Dq, *n* = 8/2; *Slitrk2*-cKO+hM3Dq, *n* = 10/2, where *n* denotes the number of cells/mice). **(F** and **G)** Whole-cell recordings of sEPSCs from CA1 pyramidal neurons in control (dark-gray), *Slitrk2*$^{V89M}$-cKI (lime), control+hM3Dq (light-gray), *Slitrk2*$^{V89M}$-cKI+hM3Dq (dark-lime) hippocampal slices. Representative traces (F) and averages of sEPSC frequency and amplitude (G; control, *n* = 11/2; *Slitrk2*$^{V89M}$-cKI, *n* = 12/2; control+hM3Dq, *n* = 11/2; *Slitrk2*$^{V89M}$-cKI+hM3Dq, *n* = 11/2, where *n* denotes the number of cells/mice). Data are presented as means ± SEMs (*$p < 0.05$, ***$p < 0.001$; nonparametric Kruskal–Wallis test with Dunn's *post hoc* test). Numerical data can be found in S1 Data. **S7 Fig. Measurement of paired pulse ratio in hippocampal CA1 pyramidal neurons and DG granule neurons from *Slitrk1*-cKO, *Slitrk2*-cKO and *Slitrk1/2*-dKO mice. (A–D)** Whole-cell recordings of eEPSC-PPRs at SC-CA1 synapses in control (gray), *Slitrk1*-cKO (red), and *Slitrk2*-cKO (green) hippocampal slices. Representative traces (A and C) and averages of eEPSC-PPRs (B and D; control in (B), *n* = 17/5; control in (D), *n* = 19/5; *Slitrk1*-cKO, *n* = 13/4; *Slitrk2*-cKO, *n* = 20/5, where *n* denotes the number of cells/mice). **(E–H)** Same as (A–D), except that whole-cell recordings of eEPSCs at TA-CA1 synapses were performed. Representative traces (E and G) and averages of eEPSC-PPRs (F and H; control in (F), *n* = 15/5; control in (H), *n* = 12/4; *Slitrk1*-cKO, *n* = 12/4; *Slitrk2*-cKO, *n* = 12/4). **(I–L)** Same as (A–D), except that whole-cell recordings of eEPSC-PPRs at MPP-DG synapses were performed. Representative traces (I and K) and averages of eEPSC-PPRs (J and L; n denotes the number of cells/mice; control in (J), *n* = 20/5; control in (L), *n* = 23/5; *Slitrk1*-cKO, *n* = 21/5; *Slitrk2*-cKO, *n* = 19/5). **(M–P)** Same as (A–D), except that whole-cell recordings of eEPSC-PPRs at LPP-DG synapses were performed. Representative traces (M and O) and averages of eEPSC-PPRs (N and P; control in (N), *n* = 20/5; control in (P), *n* = 20/5; *Slitrk1*-cKO, *n* = 21/5; *Slitrk2*-cKO, *n* = 16/5). Data are presented as means ± SEMs (two-tailed nonparametric Mann–Whitney *U* test). **(Q** and **R)** Whole-cell recordings of eEPSC-PPRs at SC-CA1 synapses of control (gray) and *Slitrk1/2*-dKO (light orange) mice. Representative traces (Q) and averages of eEPSC-PPRs (R; control, *n* = 13/4; *Slitrk1/2*-dKO, *n* = 15/3). Data are presented as means ± SEMs (two-tailed nonparametric Mann–Whitney *U* test). **(S** and **T)** Same as (Q and R), except that whole-cell recordings of eEPSC-PPRs at TA-CA1 synapses were performed.

Representative traces (S) and averages of eEPSC-PPRs (T; control, $n = 12/3$; *Slitrk1/2*-dKO, $n = 15/3$). Data are presented as means ± SEMs (two-tailed nonparametric Mann–Whitney *U* test). Numerical data can be found in S1 Data. **S8 Fig. Immunohistochemical analyses showing measured changes in excitatory synapse density of hippocampal CA1 and DG from adult *Slitrk*-cKO mice. (A** and **B)** Representative images (A) and summary graphs (B) of immunohistochemistry results for excitatory synapses in the hippocampal CA1 and DG layers of adult male control (ΔCre) and *Slitrk1*-cKO (Cre) mice. Scale bar: 20 μm. Data are presented as means ± SEMs ('*n*' denotes number of mice; SO: ΔCre, $n = 6$, Cre, $n = 6$; SR: ΔCre, $n = 6$, Cre, $n = 5$; SLM: ΔCre, $n = 6$, Cre, $n = 6$; MOL: ΔCre, $n = 6$, Cre, $n = 6$; **$p < 0.01$; two-tailed nonparametric Mann–Whitney *U* test). **(C** and **D)** Same as (A and B), except that adult male *Slitrk2*-cKO mice were analyzed. Scale bar: 20 μm. Data are presented as means ± SEMs ('*n*' denotes number of mice; SO, ΔCre, $n = 6$, Cre, $n = 6$; SR, ΔCre, $n = 5$, Cre, $n = 6$; SLM, ΔCre, $n = 6$, Cre, $n = 5$; MOL, ΔCre, $n = 6$, Cre, $n = 5$; **$p < 0.01$; two-tailed nonparametric Mann–Whitney *U* test). Numerical data can be found in S1 Data. **S9 Fig. Measurement of miniature excitatory postsynaptic currents in hippocampal CA1 pyramidal neurons from adult *Slitrk1*-cKO mice. (A** and **B)** Whole-cell recordings of mEPSCs from CA1 pyramidal neurons in control (gray), *Slitrk1*-cKO (red) hippocampal slices. Representative traces (A) and averages of mEPSCs (B; control, $n = 12/2$; *Slitrk1*-cKO, $n = 11/2$, where '*n*' denotes number of cells/mice). **(C** and **D)** Whole-cell recordings of mEPSCs from CA1 pyramidal neurons in control (gray), *Slitrk2*-cKO (green) hippocampal slices. Representative traces (C) and averages of mEPSCs (D; control, $n = 12/3$; *Slitrk2*-cKO, $n = 10/3$, where '*n*' denotes number of cells/mice). Data are presented as means ± SEMs (*$p < 0.05$; two-tailed nonparametric Mann–Whitney *U* test). Numerical data can be found in S1 Data. **S10 Fig. Deletion of Slitrk1 or Slitrk2 does not alter the ultrastructure of synapses at hippocampal CA1 and DG circuitries. (A)** Representative electron micrographic images (left) used for ultrastructure analysis of synapses at high magnification. Each component within the synapse was segmented (right). Scale bars, 100 nm. **(B)** Distance map containing information (left) about the distance from an active zone (AZ). Vesicle locations marked on the distance map (right). Scale bars, 100 nm (applies to all images). **(C–F)** Quantification of AZ length (left; cumulative distribution plot and bar graph), length of postsynaptic density (PSD) (middle; cumulative distribution plot and bar graph), and width of synaptic cleft (right) at the SC-CA1 (C), TA-CA1 (D), MPP-DG (E) and LPP-DG (F) synapses of *Slitrk1*-cKO compared with control mice. **(G–J)** Quantification of AZ length (left), PSD length (middle), and width of synaptic cleft (right) at the SC-CA1 (G), TA-CA1 (H), MPP-DG (I) and LPP-DG (J) synapses of *Slitrk2*-cKO compared with control mice. Data represents means ± SEMs (control, $n = 45/3$; *Slitrk1*-cKO or *Slitrk2*-cKO, $n = 60/4$, where $n$ denotes the number of synapses/mice; *$p < 0.05$; n.s., not significant; Kolmogorov–Smirnov test). Numerical data can be found in S1 Data. **S11 Fig. Simultaneous deletion of Slitrk1 and Slitrk2 produces phenotypes induced by ablating each Slitrk paralog from microcircuits of the hippocampal CA1 and DG regions. (A)** Schematic showing AAV injections into the hippocampal CA1 or DG of *Slitrk1/2*$^{f/f}$ mice. Electrophysiological recordings were performed 2 weeks after injections. **(B** and **C)** Whole-cell recordings of sEPSCs from CA1 pyramidal neurons of control (gray) and *Slitrk1/2*-dKO (light orange) hippocampal slices. Representative traces (B) and averages of eEPSCs (C; control, $n = 13/3$; *Slitrk1/2*-dKO, $n = 11/3$, where '*n*' denotes number of cells/mice). **(D** and **E)** Whole-cell recordings of sEPSCs from DG granule neurons of control and *Slitrk1/2*-dKO hippocampal slices. Representative traces (D) and averages of eEPSCs (E; control, $n = 12/3$; *Slitrk1/2*-dKO, $n = 12/3$). **(F** and **G)** Whole-cell recordings of eEPSCs at SC-CA1 synapses of control and *Slitrk1/2*-dKO hippocampal slices. Representative traces (F) and averages of eEPSCs (**G**; control, $n = 12/3$; *Slitrk1/2*-dKO, $n = 12/3$). **(H)** Representative traces of NMDAR/AMPAR-EPSCs at SC-CA1 synapses of control and *Slitrk1/2*-dKO hippocampal slices (left) and averages of NMDAR/AMPAR-EPSCs (right; control, $n = 12/3$; *Slitrk1/2*-dKO, $n = 12/2$). **(I** and **J)** Whole-cell recordings of aEPSCs at SC-CA1 synapses of control and *Slitrk1/2*-dKO hippocampal slices. Representative traces (I) and averages of aEPSCs (J; control, $n = 14/5$; *Slitrk1/2*-dKO, $n = 13/5$). **(K** and **L)** Representative traces of eEPSCs at TA-CA1 synapses of control and *Slitrk1/2*-dKO hippocampal slices (K) and averages of NMDAR/AMPAR-EPSCs (L; control, $n = 15/3$; *Slitrk1/2*-dKO, $n = 11/3$). **(M)** Representative traces of NMDAR/AMPAR-EPSCs at TA-CA1 synapses of control and *Slitrk1/2*-dKO hippocampal slices (left) and averages of NMDAR/AMPAR-EPSCs (right; control, $n = 15/3$; *Slitrk1/2*-dKO, $n = 11/3$). **(N** and **O)** Whole-cell recordings of

aEPSCs at TA-CA1 synapses of control and *Slitrk1/2*-dKO hippocampal slice. Representative traces (N) and averages of aEPSCs (O; control, $n = 14/5$; *Slitrk1/2*-dKO, $n = 13/5$). **(P and Q)** Whole-cell recordings of eEPSCs at MPP-DG synapses of control and *Slitrk1/2*-dKO hippocampal slices. Representative traces (P) and averages of eEPSCs (Q; control, $n = 18/5$; *Slitrk1/2*-dKO, $n = 18/5$). (R) Representative traces of NMDAR/AMPAR-EPSCs at MPP-DG synapses of control and *Slitrk1/2*-dKO hippocampal slices (left) and averages of NMDAR/AMPAR-EPSCs (right; control, n = 18/5; *Slitrk1/2*-dKO, $n = 18/5$). **(S and T)** Whole-cell recordings of aEPSCs at MPP-DG synapses of control and *Slitrk1/2*-dKO hippocampal slices. Representative traces (S) and averages of aEPSCs (T; control, $n = 14/5$; *Slitrk1/2*-dKO, $n = 13/5$). **(U and V)** Representative traces of eEPSCs at LPP-DG synapses of control and *Slitrk1/2*-dKO hippocampal slices (U) and averages of NMDAR/AMPAR-EPSCs (V; control, $n = 18/5$; *Slitrk1/2*-dKO, $n = 18/5$). **(W)** Representative traces of NMDAR/AMPAR-EPSCs at LPP-DG synapses of control and *Slitrk1/2*-dKO hippocampal slices (left) and averages of NMDAR/AMPAR-EPSCs (right; control, $n = 18/5$; *Slitrk1/2*-dKO, $n = 18/5$). **(X and Y)** Whole-cell recordings of aEPSCs at LPP-DG synapses of control and *Slitrk1/2*-dKO hippocampal slice. Representative traces (X) and averages of aEPSCs (Y; control, $n = 14/5$; *Slitrk1/2*-dKO, $n = 13/5$). Data are presented as means ± SEMs (\*$p < 0.05$, \*\*$p < 0.01$, \*\*\*$p < 0.001$; two-tailed nonparametric Mann–Whitney *U* test). Numerical data can be found in S1 Data. **S12 Fig. Characterization of SLITRK expression vectors used in the current study. (A)** Diagrams illustrating HA-tagged SLITRK variants (SLITRK1 WT, SLITRK1 PBM, SLITRK2 WT, SLITRK2 PBM, SLITRK2 V89M, and SLITRK2 ΔPDZ) used in the molecular replacement experiments. **(B)** Immunoblot analyses monitoring the expression levels of the indicated SLITRK1 (left) and SLITRK2 (right) variants in HEK293T cells. **(C)** Representative images of transfected HEK293T cells showing the expression and surface transport activity of SLITRK variants. Transfected cells were fixed (not permeabilized) and incubated with anti-HA antibody to detect the extracellular region of SLITRK (red). The intracellular level of SLITRK variants was determined by incubation with an anti-HA antibody (green) after permeabilization. Surf., surface; Intra., intracellular. Scale bar, 10 μm (applies to all images). **(D)** Quantification of surface transport activity of SLITRK variants. Data are presented as means ± SEMs ($n = 6$ images/ group; nonparametric Kruskal–Wallis test with Dunn's *post hoc* test). **(E and F)** Cell surface binding assays. Representative images (E) and summary graphs (F) showing HEK293T cells expressing N-terminally HA-tagged SLITRK variants incubated with purified Ig-fused PTPσ (Ig-PTPσ) or IgC alone (control) and analyzed by immunofluorescence imaging for Ig-fusion proteins (red) and HA (green). Data are presented as means ± SEMs ($n = 38–76$ cells/group). Scale bar, 10 μm (applies to all images). **(G)** Coomassie-stained gel of recombinant IgC and IgPTPσ used for cell-surface binding assays. The band denoted by an asterisk is likely a degradation product of the full-length PTPσ Ig-fusion proteins. Numerical data can be found in S1 Data. **S13 Fig. Slitrk2 variants used in the current study exhibit comparable surface expression in the excitatory synapses of cultured hippocampal neurons. (A)** Representative images showing the distribution of HA-tagged Slitrk2 WT and its variants (Slitrk2 PBM, Slitrk2 V89M, and Slitrk2 ΔPDZ) in cultured hippocampal neurons. Neurons were transfected at DIV7 and analyzed at DIV14 by immunofluorescence staining using anti-EGFP (blue), anti-HA (red), and anti-SHANK (green) antibodies. Scale bar, 10 μm (applies to all images). **(B)** Density quantification of HA⁺SHANK⁺ puncta. Data are presented as means ± SEMs ($n = 10$ cells/group). Abbreviation: sHA, surface HA. Numerical data can be found in S1 Data. **S14 Fig. TrkB is required to regulate evoked synchronous and asynchronous synaptic transmission in Schaffer collateral excitatory synapses of hippocampal CA1 pyramidal neurons. (A)** Schematic showing injections of the indicated AAVs into the hippocampal CA1 or DG of *Ntrk2*^f/f mice. Electrophysiological recordings were performed 2 weeks after injections to monitor alterations in spontaneous, synchronous and asynchronous synaptic transmission. **(B)** Representative immunoblotting image and quantitative analyses of the level of TrkB protein in hippocampal lysates infected with AAVs expressing Cre recombinase. **(C and D)** Whole-cell recordings of sEPSCs from CA1 pyramidal neurons of control (gray) and *Ntrk2*-cKO (blue) hippocampal slices. Representative traces (C) averages of sEPSCs (D; control, $n = 11/2$; *Ntrk2*-cKO, $n = 11/2$, where 'n' denotes the number of cells/mice). **(E and F)** Whole-cell recordings of aEPSCs at SC-CA1 synapses of control and *Ntrk2*-cKO hippocampal slices. Representative traces (E) and averages of aEPSCs (F; control, $n = 13/3$; *Ntrk2*-cKO, $n = 10/3$). **(G–K)** Whole-cell recordings of eEPSCs at SC-CA1 synapses of control

and *Ntrk2*-cKO hippocampal slices. Representative traces (G) and averages of eEPSCs (H and I; control, *n* = 11/3; *Ntrk2*-cKO, *n* = 11/3). Representative traces of NMDAR/AMPAR-EPSC at SC-CA1 synapses of control and *Ntrk2*-cKO hippocampal slices (J) and averages of NMDAR/AMPAR-EPSCs (K; control, *n* = 11/3; *Ntrk2*-cKO, *n* = 11/3). **(L–P)** Whole-cell recordings of eEPSCs at MPP-DG synapses of control and *Ntrk2*-cKO hippocampal slices. Representative traces (L) and averages of eEPSCs (M and N; control, *n* = 12/4; *Ntrk2*-cKO, *n* = 12/4). Representative traces of NMDAR/AMPAR-EPSCs at MPP-DG synapses of control and *Ntrk2*-cKO hippocampal slices (O) and averages of NMDAR/AMPAR-EPSCs (P; control, *n* = 12/4; *Ntrk2*-cKO, *n* = 12/4). **(Q–U)** Whole-cell recordings of eEPSCs at LPP-DG synapses of control and *Ntrk2*-cKO hippocampal slices. Representative traces (Q) and averages of eEPSCs (R and S; control, *n* = 12/4; *Ntrk2*-cKO, *n* = 12/4). Representative traces of NMDAR/AMPAR-EPSC at LPP-DG synapses of control and *Ntrk2*-cKO hippocampal slices (T) and averages of NMDAR/AMPAR-EPSCs (U; control, *n* = 12/4; *Ntrk2*-cKO, *n* = 12/4). Data are presented as means ± SEMs (*$p < 0.05$, **$p < 0.01$, ***$p < 0.001$; two-tailed nonparametric Mann–Whitney *U* test). Numerical data can be found in S1 Data. **S15. Fig Substituting valine for methionine at residue 89 or deleting the C-terminal PDZ domain-binding sequence in SLITRK2 does not affect its interaction with TrkB. (A** and **B)** Coimmunoprecipation experiment showing that SLITRK2 V89M and SLITRK2 ΔPDZ each interact with TrkB, comparable to SLITRK2 WT. HEK293T cells were transfected with HA-tagged SLITRK2 variant alone or together with untagged TrkB construct, followed by coimmunoprecipitation of SLITRK2 with TrkB. (A) Representative immunoblot visualized by ECL; and (B) quantitative bar graphs of coimmunoprecipitation efficiency. Data are presented as means ± SEMs (*n* = 3 independent experiments; nonparametric ruskal–Wallis test with Dunn's *post hoc* test). Input, 5%. **(C)** Knock-in strategy used to generate *Slitrk2*^V89M^-cKI mice and its validation. One loxP site each was inserted upstream of exon 2 (E2) and downstream of the 3' UTR of the murine *Slitrk2* gene and the p.V89M mutation was introduced into exon 3 (E3) by overlap extension PCR. Black arrows indicate forward and reverse primers used for genotyping. Genomic DNA was prepared from hippocampal CA1 lysates injected with AAV-Cre or AAV-ΔCre, followed by PCR amplification using the indicated primers (F, forward; R, reverse). Cre recombinase deleted E2 and E3, causing constitutive expression of *Slitrk2* p.V89M mRNA. **(D)** PCR genotyping of *Slitrk2*^V89M^-floxed mice. The band size for the *Slitrk2*^V89M^-floxed allele was 249 bp. Numerical data can be found in S1 Data. **S16 Fig. Measurement of *Slitrk* mRNA levels in adult *Slitrk2*^V89M^-cKI mice.** Quantitative RT-PCR analyses measuring mRNA levels of Slitrk paralogs in hippocampal tissues from adult male *Slitrk2*^V89M^-cKI mice. qRT-PCRs were performed using Slitrk paralog-specific probes. Gene expression levels were normalized with respect to a housekeeping gene (GAPDH). Data are presented as means ± SEMs ('*n*' denote number of mice; ΔCre, *n* = 3; Cre, *n* = 3; two-tailed nonparametric Mann–Whitney *U* test). Numerical data can be found in S1 Data. **S17. Fig Model for actions of postsynaptic Slitrk1 and Slitrk2 in mediating specification of excitatory synaptic properties in the hippocampal circuits via distinct extracellular and intracellular mechanisms. S18 Fig. Uncropped scanned images of representative immunoblots and agarose gels in the current study. S1 Table. Summary of electrophysiological phenotypes from other knockout models that depict specification of excitatory synaptic properties. S2 Table. Comparison of electrophysiological phenotypes from two different Slitrk1 loss-of-function approaches. S3 Table. Summary of ultrastructural phenotypes from the current study. S4 Table. Summary of electrophysiological phenotypes from the current study.** (DOCX)

**S1 Data. Numerical data related to Figs 1–7, S1, S2, S4, S5, S6, S7, S8, S9, S10, S11, S12, S13, S14, S15, and S16.** (XLSX)

## Acknowledgments

We are grateful to Jinha Kim (DGIST, Korea) for technical assistance and Dr. Won Do Heo (KAIST, Korea) for providing the *Ntrk2*-cKO mice.

## Author contributions

**Conceptualization:** Ji Won Um, Jaewon Ko.

**Data curation:** Dongwook Kim, Byeongchan Kim, Jinhu Kim, Na-Young Seo.

**Formal analysis:** Dongwook Kim, Byeongchan Kim, Jinhu Kim, Na-Young Seo, Chang Ho Sohn, Kea Joo Lee.

**Funding acquisition:** Jaewon Ko.

**Investigation:** Dongwook Kim, Byeongchan Kim, Jinhu Kim, Na-Young Seo, Hyeonho Kim, Kyung Ah Han, Jubeen Yoon, Christian P. Macks, Chang Ho Sohn, Kea Joo Lee.

**Methodology:** Dongwook Kim, Byeongchan Kim, Jinhu Kim, Na-Young Seo, Hyeonho Kim, Jubeen Yoon, Christian P. Macks, Chang Ho Sohn, Kea Joo Lee.

**Project administration:** Jaewon Ko.

**Resources:** Kyung Ah Han, Joris de Wit, Jaewon Ko.

**Supervision:** Ji Won Um, Jaewon Ko.

**Validation:** Dongwook Kim, Jinhu Kim, Ji Won Um.

**Visualization:** Jinhu Kim.

**Writing – original draft:** Jaewon Ko.

**Writing – review & editing:** Jaewon Ko.

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
