## [Editor Report · Decision Letter 0]

23 Jan 2025

Dear Dr Ko,

Thank you for submitting your manuscript entitled "Slitrk paralogs configure excitatory synaptic specificity via distinct extracellular and intracellular mechanisms" for consideration as a Research Article by PLOS Biology.

Your manuscript has now been evaluated by the PLOS Biology editorial staff, as well as by an academic editor with relevant expertise, and I am writing to let you know that we would like to send your submission out for external peer review.

Once your full submission is complete, your paper will undergo a series of checks in preparation for peer review. After your manuscript has passed the checks it will be sent out for review. To provide the metadata for your submission, please Login to Editorial Manager (https://www.editorialmanager.com/pbiology) within two working days, i.e. by Jan 25 2025 11:59PM.

Kind regards,

Taylor

Taylor Hart, PhD,

Associate Editor

PLOS Biology

thart@plos.org

---

## [Decision Letter · Decision Letter 1]

14 Mar 2025

Dear Dr Ko,

Thank you for your patience while your manuscript "Slitrk paralogs configure excitatory synaptic specificity via distinct extracellular and intracellular mechanisms" was peer-reviewed at PLOS Biology. It has now been evaluated by the PLOS Biology editors, an Academic Editor with relevant expertise, and by several independent reviewers.

In light of the reviews, which you will find at the end of this email, we would like to invite you to revise the work to thoroughly address the reviewers' reports.

As you will see, the reviewers find the study to be technically impressive and potentially important, despite the absence of a large conceptual advance. However, all reviewers raised concerns about many of the experiments lacking important controls.

We would like to invite a Major Revision of this submission. Please note that the revised manuscript should include data from additional control experiments in the mutant mice, to better support the reported findings. Please carefully consider the concerns raised by the reviewers and respond to them thoroughly. In addition, the revised text should avoid overstating claims.

Given the extent of revision needed, we cannot make a decision about publication until we have seen the revised manuscript and your response to the reviewers' comments. Your revised manuscript is likely to be sent for further evaluation by all or a subset of the reviewers.

**IMPORTANT - SUBMITTING YOUR REVISION**

*Re-submission Checklist*

*Published Peer Review*

*PLOS Data Policy*

*Blot and Gel Data Policy*

Sincerely,

Taylor

Taylor Hart, PhD,

Associate Editor

PLOS Biology

thart@plos.org

REVIEWS:

Reviewer #1: Functional differences of paralogs of synaptic cell adhesion molecules and their circuit-dependent roles remain incompletely understood. Focusing on the LRR superfamily members Slitrk1 and Slitrk2, Kim and colleagues use cKO mice for their analyses. They apply immunohistochemistry to demonstrate that Slitrk1 and Slitrk2 have distinct expression patterns across the adult mouse hippocampus. Slice electrophysiology and TEM demonstrate that loss of Slitrk1 or Slitrk2 in hippocampal neurons affects their intrinsic excitability, synaptic transmission, and synaptic vesicle density in a circuit-dependent manner. The authors then use slice electrophysiology to demonstrate that extra- and intra-cellular Slitrk1 and Slitrk2 interaction domains differently affect synaptic transmission.

This study is a tour de force and presents extensive electrophysiology and TEM data for Slitrk1 and Slitrk2 and their mutants. The results provide evidence that these paralogs can play synapse- and circuit-specific roles. The authors also demonstrate the importance of specific binding domains of Slitrk1 and Slitrk2 for synaptic transmission. However, several results require additional controls, and in several cases, the data do not fully support some experimental conclusions and statements. This raises a comparatively large number of points, as provided below.

Major points.

1. Potential changes in synapse number in the cKO mice and Slitrk mutants need to be addressed. This is required to assess their synaptic function and interpret electrophysiological analyses. This important analysis should, at a minimum, be performed in the cKO mice and could e.g. be addressed by filling CA1 and DG neurons with dye and quantifying dendritic spine number.

2. The authors show that CA1 neurons in the Slitrk1 cKO, the Slitrk2 cKO, and the Slitrk2 V98M mutant have altered intrinsic excitability (Supp. Fig. 2). This suggests that these mutations have synapse-independent effects on the electrophysiological properties of these cells. This effect on excitability may impact synaptic transmission. The authors need to address this concern.

3. The authors state that the pathogenic Slitrk2 V89M mutant affects CA1 micro-circuits but not the DG. However, this is based on the finding that the Slitrk2 V98M mutant affects CA1 spontaneous EPSCs and SC-CA1 asynchronous EPSCs, while these parameters are not measured in DG neurons or synapses. Without analyzing the same parameters in both CA1 and DG micro-circuits, the data does not sufficiently support that the mutant plays a micro-circuit specific role.

Minor points.

4. Related to point 3, there are several other instances in the manuscript when only some of the electrophysiology analyses are shown for specific Slitrk mutants (e.g., for the PDZ mutants in Fig. 6, synchronous and asynchronous EPSCs are not analyzed for TA-CA1 synapses, and spontaneous EPSCs are not analyzed for DG granule neurons). This makes it challenging to assess cell- and circuit-specific effects, which should, at a minimum, be discussed.

5. The control baseline values for electrophysiology recordings differ in part significantly between experiments (e.g. Fig.2D vs 2F frequency, Fig. 2N vs 2P frequency, Fig. 3H vs 3J EPSC I-O slope, Fig. 5D vs 2F frequency). Can the authors explain this discrepancy?

6. Interpreting the high-resolution immunostaining data requires specificity controls for the antibodies used. The authors can use knockout controls for Slitrk1/2 immunostaining with eMAP expanded tissue to ensure antibody specificity is maintained with this technique.

7. The authors can be commended for the controls of their mutant mice (Fig. S1, Fig. S6). But to what extent are the PBM, V89M, and PDZ mutants expressed at synapses? They could consider expressing the constructs in cultured neurons and assess whether the mutant proteins are properly sorted to synapses.

8. Can the authors describe how they could precisely and separately stimulate axon tracts that run very close to each other (e.g., MPP and LPP)?

9. The Barnes Maze test was only performed with mice in which Slitrk2 V89M is expressed by CA1 neurons. Comparative data across lines may have helped to interpret circuit-specific roles better. This limitation should be mentioned.

10. For statistical analyses, are replicates considered to be each cell/synapse or each mouse? This is unclear from the methods, and there are a few places (e.g., Fig. 1J-M figure legend) where the sampling is unclear.

11. The summary tables in the supplement are very helpful. They could also include the PBM and PDZ mutant data.

12. The authors should discuss the unexpected presynaptic localization of Slitrk1/2 and how this fits in with their other data.

13. The Slitrk2 LAR-RPTP and PDZ binding domain mutants each have one electrophysiological property changed compared to baseline (LAR-RPTP: SC-CA1 synchronous EPSCs, PDZ: spontaneous EPSCs). However, the authors conclude that "LAR-RPTP engages with Slitrk1, not with Slitrk2, in regulating synaptic specificity" and "Slitrk2 binding to PDZ domain-containing proteins is required for its regulation of specific excitatory synapse properties". It is unclear why the authors have emphasized the importance of one Slitrk2 binding domain over the other when the data suggests that both are important.

14. The Results section describing the Slitrk1/2 double knockout states that "This suggests that both Slitrk1 and Slitrk2 are functionally non-redundant in regulating a variety of excitatory synaptic properties in hippocampal subfields, but operate via the same pathways in regulating basal synaptic transmission and intrinsic excitability in CA1 pyramidal neurons.". The authors argue elsewhere in the paper (Fig. 5, 6) that these paralogs operate through different binding domains and thus different pathways. This discrepancy should be resolved.

15. The Discussion states that "(…) the Slitrk2V89M mutation recapitulates the effects of Slitrk2-cKO on spontaneous and asynchronous, but not synchronous, synaptic transmission in hippocampal circuits. These results suggest that Slitrk2V89M acts as either a loss-of-function or a gain-of-function mutation, depending on the hippocampal microcircuit involved." It is unclear how these data suggest a gain-of-function role of the V89M mutation. Instead, these results may be due to the Slitrk2 V89M protein retaining some of its endogenous functional capacity. This should be discussed.

16. This study assesses the role of Slitrks in mature synapses, not in development. This is important and can be stated in the text. This line should be changed in the Discussion: "Notwithstanding these differences, our study reinforces the idea that [...] an individual protein might perform distinct roles during development."

17. The term "synapse specificity" is referred to in the title and throughout the text of the manuscript. However, this terminology is often used to refer to the specification of synapse formation between distinct neuronal populations. To avoid misunderstanding, the authors can clarify that they refer to the specification of synaptic properties.

---------

Reviewer #2: Kim et al. conducted a comprehensive investigation into the functions of Slitrk1 and Slitrk2, two synaptic cell adhesion molecules from the Slitrk family, in regulating excitatory synaptic specificity within hippocampal networks. Utilizing conditional knockout mice alongside electrophysiological recordings, high-resolution imaging, and molecular interaction analyses, the authors sought to determine the distinct, non-redundant roles of these Slitrk paralogs in synaptic function. Their results indicate that Slitrk1 and Slitrk2 influence synaptic properties through separate molecular pathways in a circuit-dependent manner. Specifically, Slitrk1 plays a key role in modulating spontaneous excitatory synaptic activity and NMDA receptor dynamics, whereas Slitrk2 is primarily involved in controlling asynchronous neurotransmitter release and synaptic response strength. The study further uncovers notable differences in their molecular interactions. While both Slitrk1 and Slitrk2 associate with LAR-RPTPs, Slitrk2 uniquely engages with PDZ domain-containing proteins and TrkB receptors, suggesting a broader role beyond trans-synaptic adhesion. These findings emphasize the specialized and non-overlapping contributions of Slitrk proteins in hippocampal synaptic organization, reinforcing the notion that cell adhesion molecules can execute highly specific functions within neural circuits.

A particularly intriguing aspect of this study is its investigation of the schizophrenia-associated Slitrk2 V89M mutation, which selectively impairs asynchronous excitatory transmission and leads to deficits in spatial reference memory. The observation that this mutation disrupts key aspects of synaptic function despite preserving surface expression, LAR-RPTP-binding activity, and synaptogenic properties suggests a pathophysiological mechanism distinct from a simple loss-of-function effect. This finding aligns with an emerging view in neuropsychiatric research that synaptic dysfunctions linked to mental disorders often manifest in circuit-specific and activity-dependent ways, rather than through a uniform reduction in synaptic density or adhesion. However, it is critical to verify that the expression levels of the mutant Slitrk2 protein remain comparable to wild-type (WT) following recombination. This type of conditional allele design can sometimes introduce unexpected changes in mRNA or protein expression, and the authors should confirm that Slitrk2 V89M is expressed at WT levels using quantitative protein and/or transcript analyses.

While the study provides valuable mechanistic insights into how Slitrk paralogs shape excitatory circuits, several aspects of the work remain unclear or require further validation. The epitope-preserving magnified analysis of proteome (eMAP) imaging, which was used to examine Slitrk1 and Slitrk2 distribution, raises some concerns regarding signal specificity and potential background noise. Given the relatively diffuse staining patterns observed, it remains uncertain whether the reported expression differences reflect true protein localization or artifacts of the technique. Additional knockout control staining is needed to ensure that the detected signals are not simply due to non-specific fluorescence. Moreover, while the authors propose that Slitrk1 and Slitrk2 differentially colocalize with VGLUT1 and PSD-95 in distinct hippocampal layers, these findings are not fully explored or rigorously quantified. Given that both proteins are known to be postsynaptic, their differential association with VGLUT1, a presynaptic marker, warrants further explanation.

Another concern is the variability in AAV-mediated Cre recombination efficiency, which could influence the extent of Slitrk deletion across animals. The study does not provide detailed quantification of viral spread or deletion efficiency, leaving open the possibility that differences in synaptic transmission may arise from incomplete recombination rather than intrinsic Slitrk function. The authors should include immunostaining or qPCR to precisely map the regions of effective gene deletion. This will ensure that the observed phenotypes are not an artifact of heterogeneous expression patterns.

The molecular interaction studies, while informative, also raise important unanswered questions. The claim that Slitrk1 and Slitrk2 require different intracellular pathways to mediate their functions is largely supported by in vitro experiments but lacks in vivo validation. Demonstrating that dominant-negative forms of LAR-RPTPs or TrkB can disrupt Slitrk-dependent synaptic effects in hippocampal neurons would provide stronger evidence that these pathways are functionally relevant in intact circuits. However, given the complexity of this approach, this is a recommendation but not binding on the authors to perform.

Overall, this is a data-rich study based on mostly stringent and elegant experiments that appear to have been conducted carefully. The information on Slitrks provided is clearly important and of interest in the field of synapse biology. However, the study has limited conceptually advancement, insofar as the molecular mechanisms are concerned. Despite these limitations, the study makes a strong case for the functional diversification of Slitrk paralogs within hippocampal circuits. It underscores the importance of cell adhesion molecules in defining microcircuit properties, an area that remains underexplored in synaptic biology.

-----------

Reviewer #3:

In this manuscript, Kim et al. showed that two similar synaptic CAMs are involved in different aspects of the specific wiring of the hippocampus, demonstrating that these CAMs are not redundant. Using conditional knockouts for Sltrk1 and Slitrk2, the authors show that synaptic specificity emerges in a context-dependent manner, likely depending on the proximity to different partners and intracellular signalling pathways. They also showed some links to schizophrenia. There is a manuscript with a very similar structure and partially similar findings showing the involvement of Slitrk1 in synaptic specificity (Schroeder et al., Neuron 2018). This paper further confirms those findings using a conditional knockout (the previous paper used a shRNA strategy) and compares them with another member of the same family (Slitrk2). Also, this paper adds a comparison with a second region, the dentate gyrus and shows that it is context-dependent. It is not full of novelty. However, the data is solid, and the amount of work provided by the authors is substantial, demonstrating additional roles for Slitrk2 and a link to neurodevelopmental disorders. There are some points that the authors need to address to strengthen the manuscript.

Figure 1:

-What is the somatic expression of Slitrk1 and Slitrk2? Are the same neurons expressing both? The authors should run single-molecule in situ hybridization that can be quantified (assuming that the antibody staining will localize the protein at the synaptic terminals, not the soma). It is essential to consider whether different enrichments can explain part of the different functions.

-The expression of Slitrk1 in Figure 1K and 1 J is challenging to visualize. Would it be possible to improve the signal (e.g. increase brightness, and of course, at equal levels on Slitrk2)

It is difficult to conclude the quantifications in Figures 1L and 1M. Are Slitrk1 and 2 either presynaptic or postsynaptic proteins? A schema in the figure to the findings may help. Also, it will be more informative to identify the synapses by colocalising VGLUT1 and PSD95 and then co-localizing with Slitrk1 and 2, with this method or another, to confirm the localization at the synapses.

Figure 4:

-Since the infected cells are not identified in the electron microscopy experiments, it will be important to show the levels of infection the authors reach with the viral experiments. The ratio of infected cells should be provided.

- The count of docked vesicles in Slitrk1 conditional mutants decreases (Figure 4B and C). However, the frequency of sEPSCs increases (Figure 2C and D). How can the authors explain this? It seems counterintuitive, given that the sEPSCs frequency usually reflects the number of synapses and release probability. What is the density of synapses in pyramidal cells in these experiments?

Supplementary Figures:

-To assess redundancy, the authors conduct electrophysiology in the double Slitrk1cKO;Slitrk2cKO in the dentate gyrus (in Fig. S5). Is there any reason why they do not run the experiments in CA1? If any, this is more relevant since it is the core area in the manuscript; the dentate gyrus is used to show other regions.

-In Fig. S1F, Slitrk1fl/fl/Slitrk2f/f double cKO show a reduction in Gephyrin and PSD95 protein levels. The single cKO does not have a reduction in these proteins. Does this reflect a reduction in excitatory and inhibitory synapses in this model? Could the authors explain this to you?

Minor

-Page 15 "stsynaptic"

---

## [Decision Letter · Decision Letter 2]

21 Oct 2025

Dear Dr Ko,

Thank you for your patience while we considered your revised manuscript "Excitatory Slitrk paralogs configure synaptic specificity via distinct extracellular and intracellular mechanisms" for publication as a Research Article at PLOS Biology. Your revised study has been evaluated by the PLOS Biology editors, the Academic Editor, and the original reviewers.

In light of the reviews, which you will find at the end of this email, we would like to invite you to revise the work to thoroughly address the reviewers' reports.

While two of the reviewers wrote that their concerns were addressed, Reviewer 1 outlined several remaining issues. You should revise your manuscript to thoroughly address Reviewer 1's points, especially those related to Supplementary Figure 8.

Given the extent of revision needed, we cannot make a decision about publication until we have seen the revised manuscript and your response to the reviewers' comments. Your revised manuscript is likely to be sent for further evaluation by all or a subset of the reviewers.

**IMPORTANT - SUBMITTING YOUR REVISION**

*Re-submission Checklist*

*Published Peer Review*

*PLOS Data Policy*

*Blot and Gel Data Policy*

Sincerely,

Taylor

Taylor Hart, PhD,

Associate Editor

PLOS Biology

thart@plos.org

REVIEWS:

Reviewer #1: While the authors can be commended for the DREADD experiment, which supports that the loss of Slitrks affects sEPSCs independent of their effect on intrinsic excitability, key concerns remain. These concerns apply to data in the original study and new results presented in this revision.

Major points.

1. The selective removal of data from the original manuscript that had been criticized in the original submission raises questions. This applies to the VGlut1 eMAP data, which we raised questions about in point #12. We were surprised to find these VGLUT1-Slitrk colocalization data removed from Fig. 1 without any acknowledgement or rationale provided to the reviewer, especially given the authors' response to our point #12 in our review. The Gephyrin and PSD95 blots, which reviewer #3 questioned, were also removed. It is unclear why the authors removed these data instead of keeping them in and performing additional experiments or simply stating these findings as-is in the text.

2. The synapse quantifications are a critical aspect of this work. However, the pattern of immunostaining are not compelling, and the extent of change in synapse number in Fig. S8 raises questions. Based on the representative images, VGLUT1 and PSD95 do not exhibit the punctate appearance expected for this region and do not appear to colocalize well, which is expected for these two markers of excitatory synapses. How can the authors ensure this immunostaining is suitable for synapse quantification? The lack of detail provided on synapse quantification methods is not helpful, either. Using these stainings, the authors describe an ≥80% drop in synapse number in select layers of the KO hippocampus. This is very surprising, as the deletion of most synaptic proteins results in no or only modest changes in synapse number, with alpha-Neurexins being one of many examples. It should be added that such a large drop in synapse number would not align with the more modest drop in mEPSC frequency in Fig. S9.

Minor points.

3. The use of knockout controls for the eMAP experiments is important. However, some of the representative images shown are not convincing in showing specificity.

4. When quantifying Slitrk synaptic puncta in the conditional knockouts via eMAP, the authors should include an analysis of the total number and average size of Slitrk puncta to confirm whether the total amount of Slitrks has been diminished in the conditional knockouts.

5. In reference to point #5 in our initial review, it would be beneficial for the authors to explicitly state the degree of baseline variability in the electrophysiology data in the text of the manuscript. This transparency would aid readers in assessing the findings of the manuscript.

6. In response to point #14 in our initial review of the manuscript, we noted a contradiction in the discussion of Slitrk redundancy. The authors' revised sentence still bears this contradiction: "This suggests that Slitrk1 and Slitrk2 are functionally non-redundant in regulating various excitatory synaptic properties in the hippocampal DG microcircuits, but likely operate via convergent pathways in regulating basal synaptic transmission and intrinsic excitability in CA1 pyramidal neurons, as shown by the opposite phenotypes produced by loss of each Slitrk paralog". As "operating via convergent pathways" implies molecular redundancy, it is unclear how this statement could be true.

7. The authors should be more upfront about the somewhat selective application of several experimental designs in this study. For instance, the authors should explicitly state that the Barnes Maze test and several electrophysiological analyses were performed for the CA1 circuits, but not the dentate gyrus circuit in the Slitrk2 V89M mutant. This limits the extent to which this study supports the circuit-specific effects of this mutant.

8. The authors continue to make very strong statements about their results. Several points in our review of the initial manuscript asked the authors to tone down their interpretations of this data.

Reviewer #2: The authors have addressed all my concerns satisfactorily.

Reviewer #3: The authors have addressed most of my concerns. The data is robust, and the extent of work presented by the authors is impressive, illustrating additional roles for Slitrk2 and a connection to neurodevelopmental disorders.

---

## [Decision Letter · Decision Letter 3]

2 Dec 2025

Dear Jaewon,

Thank you for your patience while we considered your revised manuscript "Excitatory Slitrk paralogs configure synaptic specificity via distinct extracellular and intracellular mechanisms" for publication as a Research Article at PLOS Biology. This revised version of your manuscript has been evaluated by the PLOS Biology editors, the Academic Editor, and one of the original reviewers.

Based on the review, we are likely to accept this manuscript for publication. We know that the peer review and revision process took longer than anticipated on this paper, and we really appreciate the effort you put into addressing the reviewers' comments. Please also make sure to address the following data and other policy-related requests.

IMPORTANT: Please ensure that your revision addresses these editorial points:

--------------

**Title:

-- We would like to tweak your title to provide a bit more information on the nature of the Slitrk proteins. Is this alternative version acceptable to you?

"Paralogs of Slitrk cell adhesion molecules configure excitatory synapse specificity via distinct cellular mechanisms"

**Financial disclosure statement:

-- Please include whether the sponsors or funders played a role in the study design, data collection and analysis, decision to publish, or preparation of the manuscript. Please also include links to the funding agencies.

**Ethics:

-- The Ethics statement needs to be a separate, independent (and the first) subheading in the Material & Methods section. Please also include the national or international origin of the guide of care that you adhered to.

**Data:

-- Thank you for including the numerical data supporting the figures. I looked at some of the values shown in Data S1 and noticed a possible inconsistency related to Fig S16. It seems unclear whether the presented values are all normalized to the mean of the deltaCre value or not. In the numerical data, the mean for the Slitrk1 deltaCre is 0.905, but this looks different to what is shown in the figure (the value appears to be close to 1.0).

-- In contrast, the Slitrk2-5 deltaCre are all extremely close to 1.0, suggesting they were normalized, and this does match what is shown in the figure. Please double-check this, and also confirm that the other numerical data are correct.

**Gels:

-- Thank you for providing uncropped images of gels in S18 Fig. We noticed that there are other gel images in additional figures, including S12BG, S14B, and S15AD. Please confirm that you have included the original, uncropped, and minimally adjusted images for all of the blots and gels. Our guidelines for how to prepare and upload this data are available here: https://journals.plos.org/plosbiology/s/figures#loc-blot-and-gel-reporting-requirements

--------------

We expect to receive your revised manuscript within two weeks.

*Published Peer Review History*

*Press*

Sincerely,

Taylor

Taylor Hart, PhD,

Associate Editor

thart@plos.org

PLOS Biology

Reviewer remarks:

Reviewer #1: The authors have addressed the remaining points to further improve their study, and are transparent about all results and project designs.

---

## [Editor Report · Decision Letter 4]

9 Dec 2025

Dear Dr Ko,

Thank you for the submission of your revised Research Article "Paralogs of Slitrk cell adhesion molecules configure excitatory synapse specificity via distinct cellular mechanisms" for publication in PLOS Biology. On behalf of my colleagues and the Academic Editor, Alberto Bacci, I am pleased to say that we can in principle accept your manuscript for publication, provided you address any remaining formatting and reporting issues. These will be detailed in an email you should receive within 2-3 business days from our colleagues in the journal operations team; no action is required from you until then. Please note that we will not be able to formally accept your manuscript and schedule it for publication until you have completed any requested changes.

PRESS

Sincerely, 

Taylor Hart

Taylor Hart, PhD,

Associate Editor

PLOS Biology

thart@plos.org